# Corollary discharge promotes a sustained motor state in a neural circuit for navigation

Ni Ji[1†], Vivek Venkatachalam[1‡], Hillary Denise Rodgers[1,2], Wesley Hung[3,4], Taizo Kawano[3,4], Christopher M Clark[2], Maria Lim[3,4], Mark J Alkema[2*], Mei Zhen[3,4*], Aravinthan DT Samuel[1*]

[1]Department of Physics and Center for Brain Science, Harvard University, Cambridge, United States; [2]Department of Neurobiology, University of Massachusetts Medical School, Worcester, United States; [3]Lunenfeld-Tanenbaum Research Institute, Mount Sinai Hospital, Toronto, Canada; [4]Departments of Molecular Genetics, and Physiology, University of Toronto, Toronto, Canada

*For correspondence:
mark.alkema@umassmed.edu (MJA);
meizhen@lunenfeld.ca (MZ);
samuel@physics.harvard.edu (ADTS)

Present address: [†]Department of Brain and Cognitive Sciences, MIT, Cambridge, United States; [‡]Department of Physics, Northeastern University, Boston, United States

Competing interests: The authors declare that no competing interests exist.

**Abstract** Animals exhibit behavioral and neural responses that persist on longer timescales than transient or fluctuating stimulus inputs. Here, we report that *Caenorhabditis elegans* uses feedback from the motor circuit to a sensory processing interneuron to sustain its motor state during thermotactic navigation. By imaging circuit activity in behaving animals, we show that a principal postsynaptic partner of the AFD thermosensory neuron, the AIY interneuron, encodes both temperature and motor state information. By optogenetic and genetic manipulation of this circuit, we demonstrate that the motor state representation in AIY is a corollary discharge signal. RIM, an interneuron that is connected with premotor interneurons, is required for this corollary discharge. Ablation of RIM eliminates the motor representation in AIY, allows thermosensory representations to reach downstream premotor interneurons, and reduces the animal's ability to sustain forward movements during thermotaxis. We propose that feedback from the motor circuit to the sensory processing circuit underlies a positive feedback mechanism to generate persistent neural activity and sustained behavioral patterns in a sensorimotor transformation.

## Introduction

Animals are able to generate behaviors that persist beyond the timescales of the inciting sensory stimuli. For example, fish continue to fixate their gaze after the onset of darkness (*Seung, 1996*; *Aksay et al., 2007*). A brief aversive stimulus can evoke prolonged escape responses in many species (*Li et al., 2006*; *Herberholz et al., 2002*). Lasting behavioral states require circuit mechanisms to turn a transient stimulus into persistent neuronal activity (*Lee and Dan, 2012*; *Major and Tank, 2004*; *Hoopfer et al., 2015*; *Inagaki et al., 2019*; *Kennedy et al., 2020*). Theoretical studies have explored roles for recurrent circuitry, in particular positive feedback, in generating persistent neural activity (*Seung, 1996*). While recurrent connections are abundant in the brain, establishing causality between recurrent circuitry, persistent neural activity, and sustained behavior states has been challenging in part due to the technical difficulties in experimentally dissecting neural dynamics across entire sensorimotor pathways.

One type of recurrent connectivity that is widely observed across phyla are neuronal projections that convey motor-related signals to brain regions for sensory processing (*Wurtz, 2018*; *Crapse and Sommer, 2008*). These signals, called corollary discharge (CD) or efference copy (EC), were first proposed (*Sperry, 1950*; *Holst and Mittelstaedt, 1950*) and subsequently demonstrated (*Poulet and Hedwig, 2002*; *Requarth and Sawtell, 2014*; *Schneider et al., 2014*; *Kim et al., 2015*) as

mechanisms to cancel sensory reafferents generated by self-motion. Recent studies, however, have also identified examples of motor-to-sensory feedback that are excitatory (*Hendricks et al., 2012*; *Lee et al., 2013*; *Zagha et al., 2013*; *Fu et al., 2014*). Emerging evidence from cortex- or brain-wide activity patterns has revealed widespread representation of motor states in sensory processing regions (*Kato et al., 2015*; *Stringer et al., 2019*; *Aimon et al., 2019*; *Musall et al., 2019*; *Marques et al., 2020*), which is likely to be caused, at least in part, by motor-to-sensory feedback. These observations suggest diverse roles for CD in sensorimotor processing.

The compact nervous system and optical accessibility of *Caenorhabditis elegans* make it possible to explore circuit mechanisms that underlie sustained behavioral states in intact animals (*Gao et al., 2015*). *C. elegans* requires persistent motor states to navigate variable sensory environments. During locomotion, the animal alternates between continuous forward movements (runs) and brief back-ward movements (reversals). When navigating through a chemical or thermal gradient, *C. elegans* employs a biased random walk strategy, selectively extending forward runs when moving along pre-ferred directions and shortening runs when veering off course (*Pierce-Shimomura et al., 1999*; *Ryu and Samuel, 2002*; *Iino and Yoshida, 2009*; *Hedgecock and Russell, 1975*; *Mori and Ohshima, 1995*; *Luo et al., 2014a*). During forward runs, *C. elegans* also gradually steers its heading angle towards the preferred direction, a strategy called klinotaxis (*Ward, 1973*; *Iino and Yoshida, 2009*). Forward runs play a crucial role in *C. elegans* navigation, but the neural circuit basis for run persistence remains poorly understood (*Ferrée and Lockery, 1999*).

Here, we study the sensorimotor pathway that controls navigation towards warmer temperatures (positive thermotaxis). Past studies have revealed a multilayered neural circuit underlying this behav-ior. AFD is the thermosensory neuron that mediates both positive and negative thermotaxis (*Luo et al., 2014a*; *Hawk et al., 2018*). Its principal chemical synaptic partner, AIY, is a first-layer interneuron specifically required for positive thermotaxis. AIY responds to temperature variations due to excitatory input from AFD (*Clark et al., 2006*; *Clark et al., 2007*; *Narayan et al., 2011*; *Hawk et al., 2018*). AIY sends synaptic outputs to multiple second-layer interneurons, which in turn synapse onto head motor neurons and premotor interneurons that drive forward runs or reversals. AIY has been shown to promote the speed and duration of forward locomotion (*Li et al., 2014*; *Tsalik and Hobert, 2003*). AIY is also postsynaptic to multiple other sensory neurons and is thought to play a role in navigation across different sensory modalities by controlling run duration (*Gray et al., 2005*; *Wakabayashi et al., 2004*; *Tsalik and Hobert, 2003*).

We probed mechanisms by which AIY biases random walks during positive thermotaxis. Imaging AIY activity in moving animals reveals that AIY encodes both temperature and motor information. Previous studies have found AIY to encode either sensory stimuli (*Chalasani et al., 2007*; *Clark et al., 2006*) or locomotory state (*Li et al., 2014*; *Luo et al., 2014b*) in different experimental paradigms. Here, we found that thermosensory response in AIY is gated by the locomotory state of the animal. In the absence of thermosensory stimuli, AIY activity reliably encodes the locomotory state. When exposed to thermal fluctuations during forward runs, AIY activity exhibits variability but tends to be excited by warming and inhibited by cooling. During reversals, AIY activity remains low and does not encode thermal stimuli.

We demonstrate that the motor state encoding in AIY represents a CD signal from premotor interneurons that drive the forward run and reversal states. This CD signal requires RIM, an interneu-ron that is connected with premotor interneurons. In the absence of RIM, AIY activity reliably enco-des thermal stimuli regardless of the locomotory state. Loss of RIM also leads to increased thermosensory representation in premotor interneurons and motor neurons. At the behavioral level, loss of RIM led to defects in positive thermotaxis by reducing the persistence of the forward run state. Our results establish a role for CD in sustaining a motor state in variable or fluctuating sensory environments.

## Results

### Forward movements are sustained across thermal fluctuations during positive thermotaxis

*C. elegans* navigates towards temperatures that correspond to prior thermal experience. To evoke positive thermotaxis, we placed young adults cultivated at 25°C on a linear thermal gradient

spanning 19–23°C (*Figure 1*). Consistent with earlier reports, these animals exhibited biased random walk and klinotaxis towards warmer temperatures (*Figure 1B*; *Luo et al., 2014a*; *Yamaguchi et al., 2018*): runs that pointed in favorable directions were lengthened (*Figure 1B*); forward heading angles gradually reoriented towards temperatures that correspond to prior experience (*Figure 1B*). Without a temperature gradient, there was no evident modulation of either run length or heading angle (*Figure 1B*).

Individual trajectories during positive thermotaxis revealed periods of forward movement that carry the animal up the temperature gradient. Although these periods of forward movement are persistent in duration, they are not always persistent in direction (*Figure 1C*, *Figure 1—figure supplement 1*). *C. elegans* experiences temporal changes in temperature on spatial gradients because of its own movements. Most runs – even those that orient the animal towards warmer temperatures – will involve periods of both warming and cooling stimuli because of frequent changes in movement direction (*Figure 1D*, *Figure 1—figure supplement 1*). Thus, *C. elegans* reveals an ability to sustain forward movement up temperature gradients despite transient cooling fluctuations.

## Thermosensory representation in AIY is gated by the motor state

To uncover circuit mechanisms for sustaining forward movement against thermal fluctuations, we simultaneously imaged the calcium activity across a group of neurons involved in thermosensory processing, locomotory control, or both. These include the principal thermosensory neuron AFD, its primary postsynaptic partner AIY, the premotor interneuron AVA, the left-right pair of the head motor neurons RME, SMDV, and SMDD, and the RIM interneuron that extensively connects with premotor interneurons and motor neurons (*Figure 2A, B*). To measure locomotory behavior while minimizing motion artifact, we performed imaging in semi-constrained animals that exhibited sinusoidal movements akin to those observed in free-moving animals (*Figure 2—figure supplement 1*, also see Materials and methods). This approach allowed us to simultaneously examine the encoding of thermosensory and motor information in the same neurons.

First, we measured the activity of the AFD thermosensory neuron and AIY, its principal postsynaptic partner, by calcium imaging in moving animals. Subjected to oscillating temperatures below the preferred temperature, *C. elegans* exhibits positive thermotaxis. As previously reported (*Clark et al., 2006*), AFD's activity phase locks to periodic variations in temperatures, rising upon warming and falling upon cooling (*Figure 2C, D*). AFD activity did not covary with transitions between forward run and reversal states (*Figure 2C*), indicating that motor commands arise downstream of the thermosensory neuron.

Next, we simultaneously monitored AIY calcium dynamics along with components of the motor circuit known to code forward and reversal motor states (*Figure 2A, B*, *Figure 2—videos 1* and *2*). We found that AIY encodes both temperature variations in a motor state-dependent manner. During forward movements, AIY's calcium activity on average increased upon warming and decreased upon cooling with substantial trial-to-trial variability (*Figure 2E–G*). During reversals, these responses were largely absent (*Figure 2E*).

Unlike AFD and AIY, all motor circuit neurons that we examined exhibited little to no phase-locked response to thermosensory stimulation. Instead, motor neurons reliably encoded the forward versus reversal movement states (*Figure 2E, F*, see the stimulus cross-correlation plots). In animals subjected to oscillating thermosensory stimulation, AVA calcium activity exhibited high and low states that correlated with backward and forward movement, respectively (*Figure 2E*). This observation is consistent with the known activity profile of AVA in the absence of thermal stimulation and its functional role in promoting the reversal state (*Chalfie et al., 1985*; *Kawano et al., 2011*; *McCormick et al., 2011*; *Kato et al., 2015*). Similarly, the RIM interneuron was selectively active during reversals and its activity positively correlated with AVA activity. The head motor neurons RME, SMDD, and SMDV were selectively active during forward runs. The SMD neurons exhibited alternating activity patterns, consistent with previous reports (*Hendricks et al., 2012*). Together, these observations indicate that, for positive thermotaxis, the sensorimotor transformation progresses through three layers of processing (*Figure 2A*): the thermosensory neuron encodes only thermal stimuli; the first layer interneuron encodes both thermal stimuli and motor states; the premotor and motor neurons primarily encode the motor state.

To elucidate which of these activity patterns are driven by thermosensory inputs, we next measured the activity of these neurons under constant temperature (*Figure 2H*). We found that the

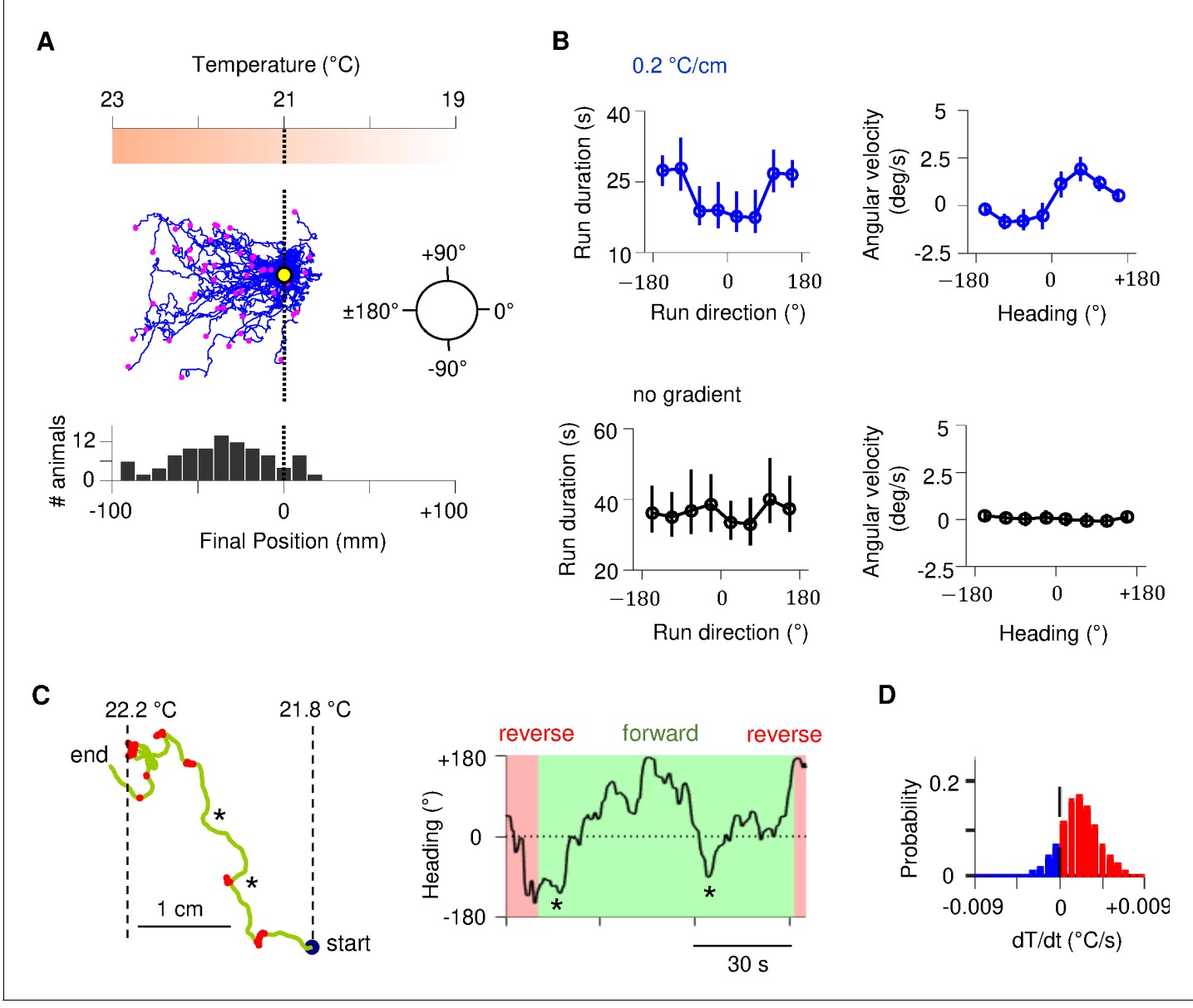

**Figure 1.** Sustained forward motor state despite temperature fluctuations during positive thermotaxis. (**A**) Example trajectories of wild-type *C. elegans* cultivated at 25°C migrating up a linear temperature gradient over 20 min. Top: schematics of the thermal gradient. Middle: trajectories of 49 animals during positive thermotaxis. The starting points of all trajectories are aligned (yellow dot) and the end points are marked by magenta dots. Bottom: a histogram of the final location of animals. (**B**) Left column: duration of forward runs as a function of their overall direction (vector pointing from the starting point to the end point of the run). Right column: instantaneous velocity during forward runs as a function of the instantaneous heading angle. Top row: data from animals exposed to spatial thermal gradients (top, N = 140). Bottom row: data from animals under constant temperature surfaces (bottom, N = 73). (**C**) Thermotaxis trajectory of a single animal during thermotaxis with alternating periods of forward movement and reversals (left), and the instantaneous heading angle over time during one extended period of forward movement within the trajectory (right). Asterisks denote periods where the heading direction pointed down the thermal gradient. (**D**) Histogram of temporal changes in temperature ($dT/dt$) experienced by animals during forward runs that ended up pointing up the temperature gradient. Data from N = 140 wild-type animals exposed to linear thermal gradient and N = 73 wild-type animals exposed to constant temperature of 21°C. Error bars are standard errors of the mean (s.e.m.).

The online version of this article includes the following source data and figure supplement(s) for figure 1:

**Source data 1.** Thermotaxis assay data.

**Figure supplement 1.** Distribution of thermal fluctuations experienced by *C. elegans* animals during positive chemotaxis.

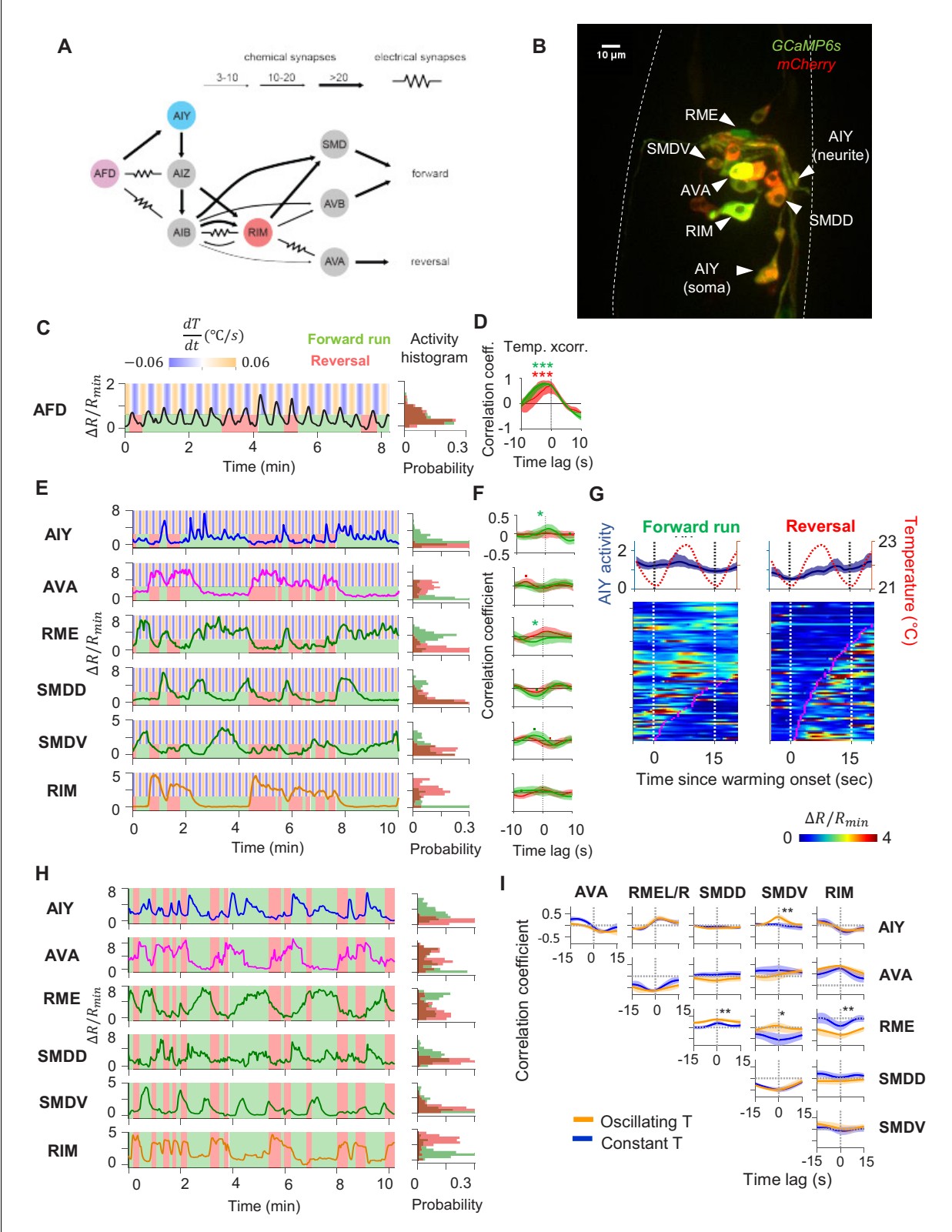

**Figure 2.** Characterization of circuit-level neural activity in the behaving animal with fluctuating or constant temperatures. (**A**) Anatomical connections among neurons implicated in positive thermotaxis, locomotory control, or both. Connectivity is inferred from both the original *C. elegans* connectome (***White et al., 1986***) and a recent reconstruction of the connectome across the developmental time course (***Witvliet et al., 2020***) (see http://www. nemanode.org) (**B**) Example maximum projection of a confocal z-stack taken from a transgenic animal expressing GCaMP6s and mCherry in neurons

*Figure 2 continued on next page*

*Figure 2 continued*

examined in this study. (C) Example ratiometric calcium activity trace (left panel) and histogram (right panel) of the thermosensory neuron AFD in response to oscillating temperature. (D) Average cross-correlation function between thermal stimuli and AFD activity during forward run (green) or reversal (red) states. N = 5 wild-type animals. (E) Simultaneously measured calcium activity of interneurons and motor neurons involved in thermosensory processing and/or motor control. The activity traces (left) and activity histograms (middle) are from the same sample dataset. (F) Average cross-correlation functions between the thermal stimuli and individual neurons shown in (E), conditioned on the animal in forward run (green) or reversal (red) states. N = 6 wild-type animals. Error bars are 95% CI of the mean. Wilcoxon rank-sum test was used to compare the peak mean cross-correlation values during forward runs (green) versus reversals (red). *p<0.05; **p<0.01; ***p<0.001; no asterisk p>0.05. (G) Thermal stimulus-triggered activity of the AIY interneuron during forward runs (left column) and reversals (right column) in wild-type animals (N = 5). Individual stimulus epochs from the same neuron under the given motor state were concatenated into heat maps, with the average calcium activity trace shown on top. (H) Simultaneous recording of the activity of interneurons and motor neurons under constant temperature. The histograms on the right are derived from the sample activity traces to the left. (I) Pairwise cross-correlation functions among neurons examined in (E) and (H) under oscillating (orange, N = 6) or constant temperature (blue, N = 7). Error bars are standard errors of the mean (s.e.m.). Wilcoxon rank-sum test was used to compare the peak mean cross-correlation values between data under oscillating temperature versus data under constant temperature. *p<0.05; **p<0.01; ***p<0.001; no asterisk p>0.05.

The online version of this article includes the following video, source data, and figure supplement(s) for figure 2:

**Source data 1.** wild type circuit activity under thermal stimulation.
**Figure supplement 1.** Behavior state annotation in free-moving and semi-constrained animals.
**Figure supplement 2.** Thermal response of AIY under different locomotory states.
**Figure 2—video 1.** Circuit-wide neural activity in semi-constrained wild-type animal exposed to oscillating temperature.
https://elifesciences.org/articles/68848#fig2video1
**Figure 2—video 2.** Circuit-wide neural activity in semi-constrained wild-type animal exposed to constant temperature.
https://elifesciences.org/articles/68848#fig2video2

activity of all six neurons continued to be modulated by locomotory state in the same way as when exposed to oscillating temperature. Furthermore, the pairwise cross-correlation between neurons remained the same under both oscillatory and constant temperature (*Figure 2I*). Specifically, AIY and RME were positively correlated with one another and were anti-correlated with AVA. The head motor neurons SMDD and SMDV were strongly anti-correlated with one another, consistent with previous reports (*Hendricks et al., 2012*). Both SMDD and SMDV were positively correlated with RME and AIY, consistent with their role in controlling head oscillations during forward locomotion (*Pirri et al., 2009*). These observations indicate that the widespread encoding of motor state in neurons downstream of AFD does not require thermosensory input and may instead reflect an intrinsic circuit property.

## Motor coding in AIY is a CD signal that requires RIM

Our finding that AIY encodes thermal information in a manner that depends on motor state suggests a critical role in sensorimotor transformations during positive thermotaxis. In animals exposed to either constant or oscillating temperatures, AIY activity consistently rises at the beginning of forward runs and decays at the onset of reversals (*Figure 3A, B*). Moreover, AIY exhibited persistent activation, the duration of which coincided reliably with that of the forward run state (*Figure 3—figure supplement 1*). How does AIY, a first-order interneuron, acquire a robust motor signal?

Because AIY has an established role in promoting forward locomotion, we first tested whether motor representation in AIY arises due to feedforward output from AIY to the downstream circuit. We imaged AIY activity after blocking vesicle release from AIY through cell-specific expression of tetanus toxin (TeTx) (*Figure 3C*). Despite the lack of synaptic and dense-core-vesicle-dependent chemical release (*Whim et al., 1997*), AIY activity remained strongly coupled to the motor state, implying that AIY must receive the motor state signal.

We explored the possibility that proprioception, elicited by movement itself, underlies the calcium response in AIY. We imaged neural activity in AIY and the rest of the thermosensory circuit in immobilized animals under constant temperature. As in moving animals, AIY's activity remained anti-correlated with neurons active during reversals (AVA) and correlated with neurons active during forward movement (RME and SMDD/V) (*Figure 3—figure supplement 2*). Furthermore, aligning AIY activity to the onset and offset of AVA activity revealed average activity patterns similar to that of

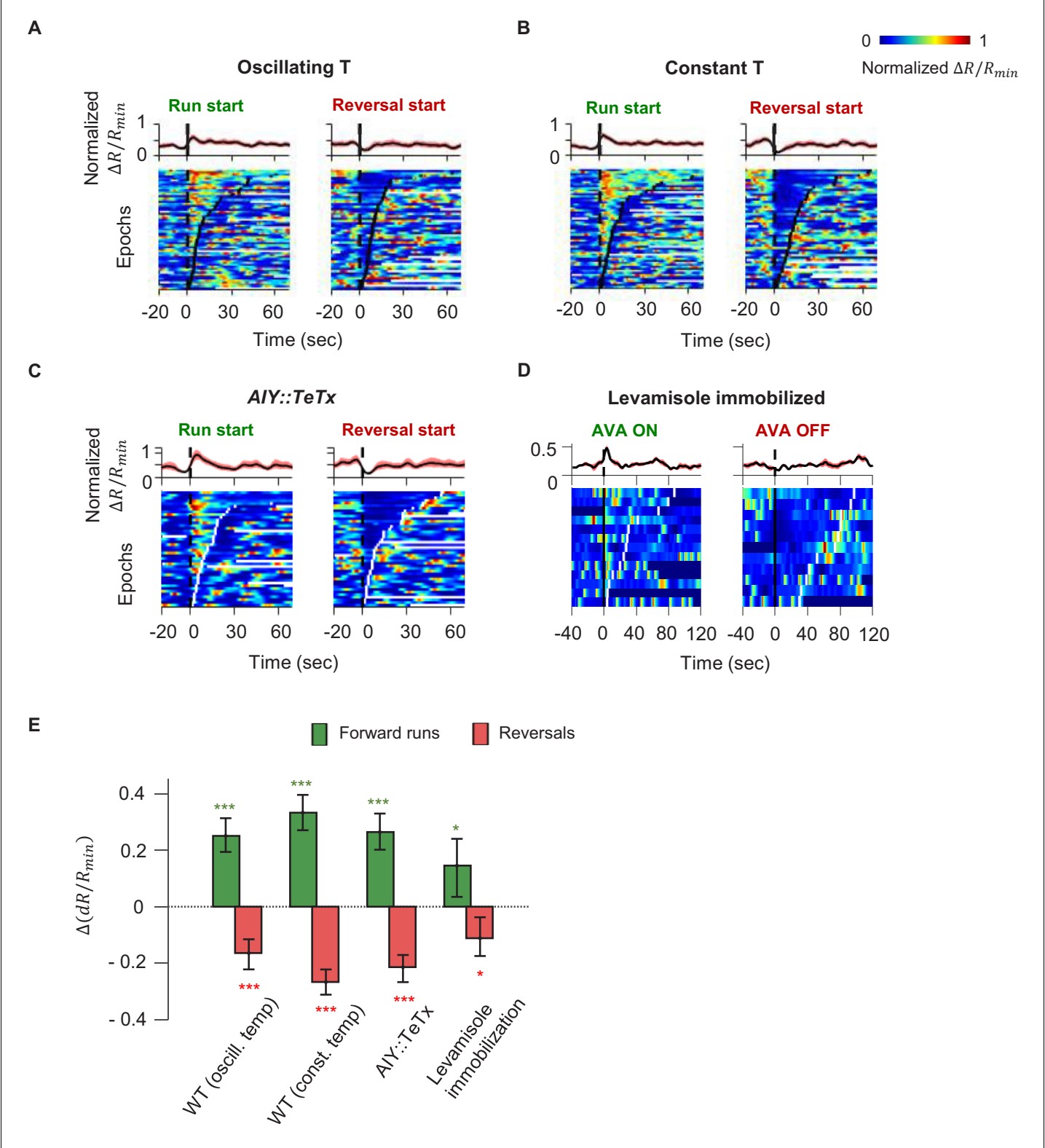

**Figure 3.** Motor-related activity in the AIY interneuron represents a corollary discharge signal. (**A, B**) Calcium activity of the AIY interneuron aligned to the onset of forward runs (left column) or reversals (right column) in animals exposed to oscillating temperature (**A**, N = 6) or constant temperature (**B**, N = 5). Each row of the heat plots represents AIY calcium activity during a single behavioral epoch. The curve on top of each panel represents activity dynamics averaged across individual epochs. Broken lines indicate the onset and offset of each behavior epochs. (**C**) Calcium activity of the AIY

*Figure 3 continued on next page*

*Figure 3 continued*

interneuron aligned to the onset of forward runs or reversals in animals expressing tetanus toxin (TeTx) specifically in AIY (N = 4). (**D**) Calcium activity of AIY aligned to onset or offset of AVA activation in animals immobilized by the cholinergic agonist levamisole (N = 4). The ON and OFF states of AVA activity are defined by binarizing AVA activity using the Otsu method. See Materials and methods for details. (**E**) Change in AIY activity before versus. after the onset of forward runs (green) or reversals (red) for datasets shown in (**A**) and (**B**). Wilcoxon Signed rank test was used to test if the change in AIY activity has a median significantly different from zero, *p<0.05; **p<0.01; ***p<0.001; n.s., non-significant.

The online version of this article includes the following source data and figure supplement(s) for figure 3:

**Source data 1.** AIY activity analysis - wild type.
**Figure supplement 1.** Quantification of persistent activity in AIY in wild-type animals.
**Figure supplement 2.** Circuit-level neural activity in immobilized animals.

the motor state representation in moving animals (*Figure 3D, E*). *C. elegans* movement is not required for AIY activity to reflect motor state, arguing against proprioception.

We next asked whether AIY receives CD from neurons that encode the motor command. We imaged AIY activity in moving animals upon ablation of AIY's downstream interneurons and premotor interneurons (*Figure 4A*). For interneurons, we focused on AIB and RIM. AIB shares electrical synapses with RIM and the AFD thermosensory neuron (*White et al., 1986*). In the context of chemotaxis, AIB and RIM have been shown to regulate variability in the neuronal and behavioral response to olfactory inputs (*Gordus et al., 2015*). We also tested AVA and AVB, premotor interneurons that regulate reversal and forward movement, respectively. Ablations were performed by expressing flavoprotein miniSOG, which induces acute functional loss and neuronal death by photoactivation (*Qi et al., 2012*).

We found that ablating AIB did not abolish the motor state representation in AIY (*Figure 4A, B*). Neither did the removal of the premotor interneurons AVA or AVB alone (*Figure 4A*, *Figure 4—figure supplement 1A, B*). However, AIY lost its motor state representation when we ablated RIM either by itself or in combination with other premotor interneurons (*Figure 4A, B*, *Figure 4—figure supplement 1C*).

RIM activity has been shown to be correlated with the AVA premotor interneuron that promotes reversals and anti-correlated with the AVB premotor interneuron that promotes forward runs (*Kawano et al., 2011*; *Gordus et al., 2015*; *Kato et al., 2015*). RIM has been shown to promote long reversals (*Gray et al., 2005*) and to suppress head oscillations during reversals (*Alkema et al., 2005*). A recent study demonstrated that RIM also promotes the stability of forward runs when hyperpolarized (*Sordillo et al., 2021*). To probe whether RIM is required for the motor state signal to appear in AIY, we optogenetically activated either AVA or AVB while simultaneously measuring AIY calcium activity in immobilized animals. Activation of AVB using the light-gated opsin chrimson triggered an increase in AIY calcium levels (*Figure 4C, D*). Activation of AVA (*Klapoetke et al., 2014*) triggered a decrease in AIY calcium levels (*Figure 4E, F*). When RIM was ablated, AIY calcium signals no longer responded to optogenetic activation of either AVA or AVB, suggesting that RIM is part of the CD pathway from the motor circuit to AIY. Without RIM, AIY activity no longer reflected or depended on the motor state, but the premotor interneurons AVA continued to encode the backward and forward movement, albeit with reduced bimodal activity (*Figure 5A, C*). Thus, RIM is not essential for generating motor commands, but is necessary to relay motor information to AIY, a first-layer interneuron.

## RIM-mediated CD does not depend on chemical synaptic transmission

We sought synaptic mechanisms by which RIM may contribute to the CD pathway. RIM expresses VGLUT3/EAT-4, indicating the potential involvement of glutamatergic synaptic transmission (*Serrano-Saiz et al., 2013*). RIM also synthesizes tyramine, a monoamine neuromodulator (*Alkema et al., 2005*). We thus imaged AIY activity in loss-of-function mutants for glutamatergic signaling (*VGLUT3/eat-4*), tyramine synthesis (*TDC/tdc-1*), vesicular monoamine transport (*VMAT/cat-1*), and peptidergic signaling (*CAPS/unc-31*). AIY activity co-varied with the motor state in all mutants, but the difference in AIY activity between the forward run and reversal states was less distinct in mutants defective for vesicular monoamine transport (*VMAT/cat-1*) or tyramine synthesis (*TDC/tdc-1*) (*Figure 4—figure supplement 2A, B*). Blocking vesicle fusion in RIM by expressing TeTx (P*tdc-1:: TeTx*) also attenuated the motor state representation in AIY (*Figure 4—figure supplement 2A, B*).

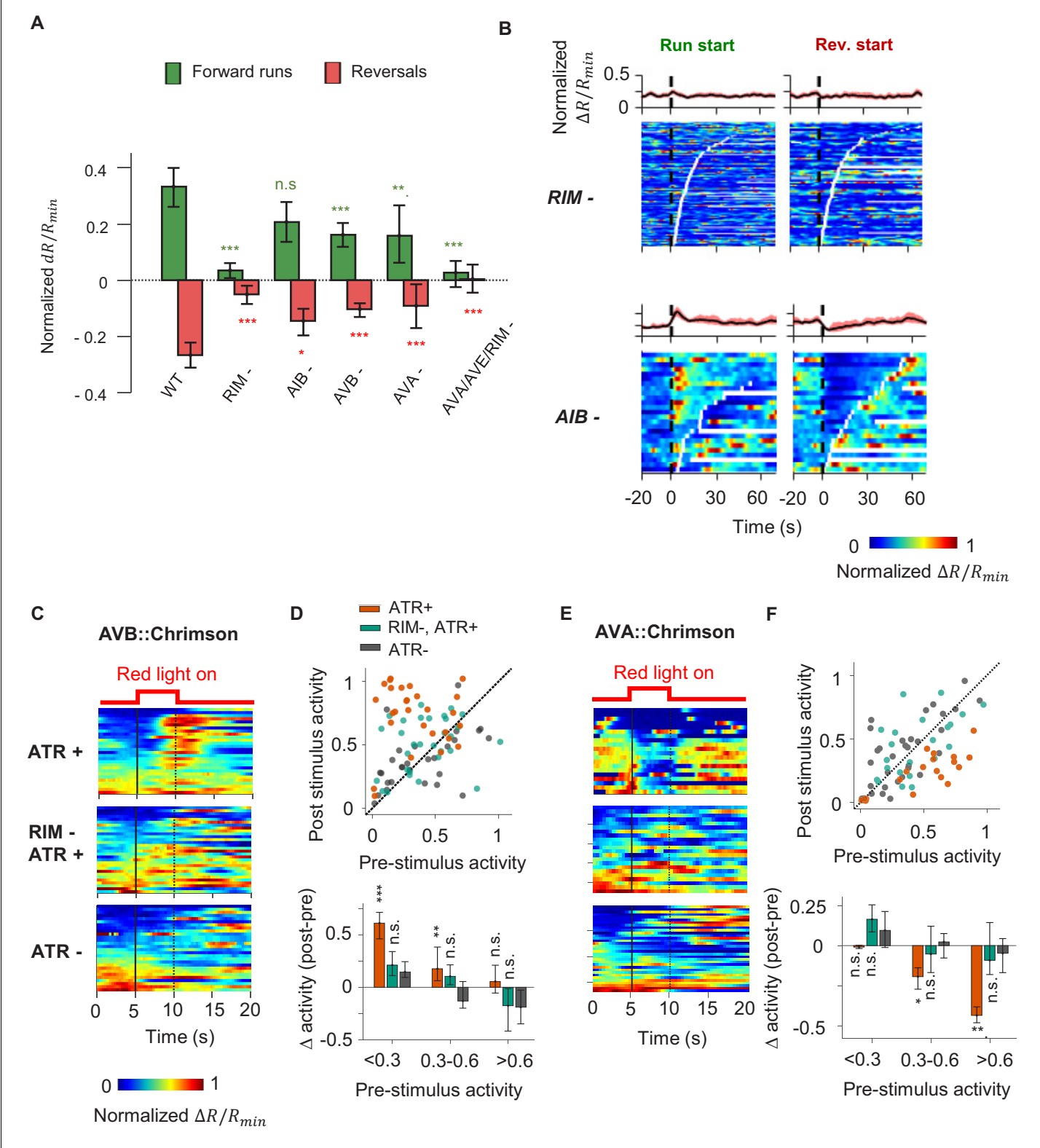

**Figure 4.** A corollary discharge (CD) pathway that requires the RIM interneuron couples AIY activity with the motor state. (**A**) Change in AIY activity before versus. after the onset of forward runs (green) or reversals (red) in animals where candidate neurons for relaying the CD signal have been ablated. Data are from RIM-ablated animals (N = 9); AIB-ablated animals (N = 3); AVB-ablated animals (N = 4); AVA-ablated animals (N = 4); and AVA/AVE/RIM (*nmr-1::miniSOG*)-ablated animals (N = 4). Error bars are 95% CI. Wilcoxon rank-sum test was used to test if the distributions of AIY activity

*Figure 4 continued on next page*

*Figure 4 continued*

before and after the onset of motor states have the same median: *p<0.05; **p<0.01; ***p<0.001; no asterisk p>0.05. (B) AIY activity aligned to the onset of forward runs (left column) or reversals (right column) in the animals where neurons RIM (upper panels, N = 9) or AIB (lower panels, N = 3) have been genetically ablated. (C) AIY activity in response to optogenetic stimulation of the AVB premotor interneurons in wild-type animals grown on all-trans retinal (ATR) (top, N = 5), RIM-ablated animals grown on ATR (N = 6), and wild-type animals grown without ATR (N = 3). (D) Top: AIY activity 2.5 s before (pre-stimulus) versus 2.5 s at the end of the AVB opto-stimulation (post-stimulus) under experimental conditions shown in (D). Bottom: average change in AIY activity pre-and post-stimulation under experimental conditions shown (D). Trials are sorted into three groups based on pre-stimulation AIY activity. Same datasets as in (C). (E) AIY activity in response to optogenetic stimulation of the AVA premotor interneurons in wild-type animals grown on ATR (top, N = 4), RIM-ablated animals grown on ATR (middle, N = 5), and wild-type animals grown without ATR (bottom, N = 5). (F) Top: AIY activity 2.5 s before (pre-stimulus) versus 2.5 s at the end of the AVA opto-stimulation (post-stimulus) under experimental conditions shown in (E). Bottom: average change in AIY activity pre- and post-stimulation under experimental conditions shown in (E). Trials are sorted into three groups based on pre-stimulation AIY activity. Same datasets as in (E). For (E) and (F), error bars are 95% CI; Wilcoxon signed-rank test was used to test if the average post-stimulation change in AIY activity was significantly different from 0. *p<0.05; **p<0.01; ***p<0.001; no asterisk p>0.05.

The online version of this article includes the following source data and figure supplement(s) for figure 4:

Source data 1. AIY activty analysis - mutant.

Figure supplement 1. AIY activity aligned to behavioral states in animals where candidate neurons for relaying the corollary discharge signal have been ablated.

Figure supplement 2. AIY activity aligned to motor states in mutant and transgenic animals defective in various modes of chemical transmission.

Since perturbation of chemical synaptic transmission did not fully abolish motor-related activity in AIY, neuronal communication that is independent of classic chemical synaptic transmission is likely involved in relaying CD to AIY. As previously reported (*Kawano et al., 2011*; *Kato et al., 2015*; *Gordus et al., 2015*), RIM activity is strongly correlated with the AVA premotor interneuron, higher during reversals and lower during forward movement. AIY, on the other hand, exhibits increased activity during forward movement (*Figure 3*). This sign reversal may be explained by an inhibitory input from RIM to AIY. Alternatively, RIM may play a permissive role in allowing the motor-related feedback to AIY. Taken together, our results suggest that the joint representation of sensory and motor signals in AIY arises from separate sources: feedforward input from AFD and feedback from the motor circuit that is dependent on RIM.

## RIM-dependent CD promotes persistent forward states and more effective thermotaxis

RIM plays a critical role in the motor state-dependent modulation of AIY calcium activity. This prompted us to examine the effect of disrupting the CD signal on sensorimotor transformations. When RIM-ablated animals were subjected to oscillating temperatures, AIY activity was no longer coupled to the motor state, but instead reliably tracked temperature fluctuations during both forward and backward movements (*Figure 5A, B*, *Figure 5—figure supplement 1A, B*, *Figure 5—video 1*). Under oscillating temperature, the average duration of AIY activation was shortened compared to wild type, though no significant difference was observed under constant temperature (*Figure 5D*).

When RIM was ablated, we were also able to detect the representation of thermosensory oscillations in the activity pattern of the AVA premotor interneuron and the head motor neurons RME and SMDV (*Figure 5A*). This observation suggests that the loss of the RIM-dependent motor signal resulted in a sensorimotor circuit that becomes more susceptible to fluctuations in thermosensory input. Without RIM and the motor state encoding in AIY, fluctuations in thermosensory inputs are readily propagated to the motor circuit. Thus, the RIM-dependent CD may play an important role in sustaining neural activity states through fluctuating sensory inputs.

We tested this hypothesis by examining the effect of RIM ablation on positive thermotaxis (*Figure 6A*, *Figure 6—figure supplement 1A*). Compared to wild-type animals, RIM-ablated animals exhibited an overall reduction in thermotaxis bias (*Figure 6B*). These animals were specifically defective in their ability to sustain forward locomotion when moving up the thermal gradient, while their ability to gradually modify heading angle during a forward run remained intact (*Figure 6C*, *Figure 6—figure supplement 1A*). At constant temperature, run durations are similar between wild-type and RIM-ablated animals (*Figure 6—figure supplement 1B*). Thus, the loss of RIM specifically disrupted the animal's ability to sustain forward runs up temperature gradients.

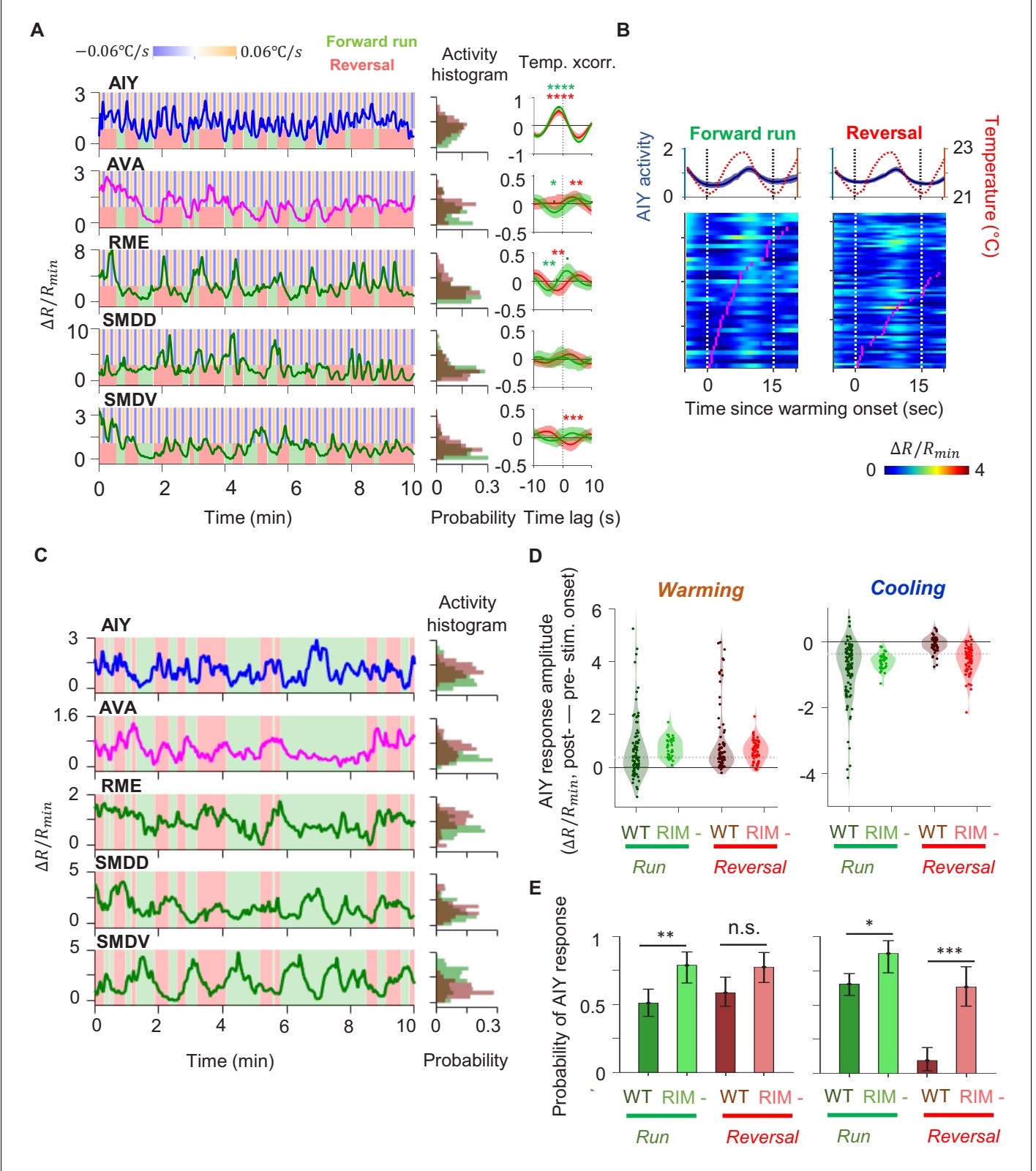

**Figure 5.** Characterization of circuit-level neural activity in behaving RIM-ablated animals under fluctuating or constant temperature. (**A**) Simultaneously measured activity of AIY and neurons of the motor circuit in RIM-ablated animals under oscillating temperature. Middle panels show histograms of neuron activity during forward runs (green) or reversals (red) for the dataset to the left. Right panels show average cross-correlograms between neural activity and thermal stimuli during forward runs and reversals across RIM-ablated animals (N = 3). Error bars are 95% CI of the mean. Wilcoxon rank-sum

*Figure 5 continued on next page*

*Figure 5 continued*

test was used to compare the peak mean cross-correlation values during forward runs (green) versus reversals (red). *p<0.05; **p<0.01; ***p<0.001; no asterisk p>0.05. (**B**) Thermal stimulus-triggered activity of the AIY interneuron during forward runs (left column) and reversals (right column) in RIM-ablated animals (N = 3). Individual stimulus epochs from the same neuron under the given motor state were concatenated into heat maps, with the average activity trace shown on top. (**C**) Simultaneously measured activity of AIY and neurons of the motor circuit in RIM-ablated animals under constant temperature (left). Right panels show histograms of neuron activity during forward runs (green) or reversals (red). (**D**) Violin plots showing the distribution of changes in AIY activity in response to warming (left) or cooling (right) stimuli under forward run or reversal state in wild-type and RIM-ablated animals. Dotted black lines indicate threshold values used to calculate the response probability in (**E**). (**E**) Probability that the magnitude of change in AIY activity upon warming or cooling is above defined thresholds in wild-type or RIM-ablated animals. See Materials and methods for details. For (**D**) and (**E**), N = 5 wild-type animals and N = 3 RIM-ablated animals. Error bars are 95% CI of the mean. Wilcoxon rank-sum test *p<0.05; **p<0.01; ***p<0.001; n.s., non-significant.

The online version of this article includes the following video, source data, and figure supplement(s) for figure 5:

**Source data 1.** Circuit activity under thermoal stimulation in RIM ablated animals.
**Figure supplement 1.** Analysis of AIY activity in RIM-ablated animals.
**Figure 5—video 1.** Circuit-wide neural activity in semi-constrained RIM-ablated animal exposed to oscillating temperature.
https://elifesciences.org/articles/68848#fig5video1

If RIM-dependent motor feedback serves to filter out transient thermal fluctuations during positive thermotaxis, then loss of RIM should render the forward run state more susceptible to thermal variations. To test this prediction, we analyzed how periods of cooling affected the forward run state in wild-type and RIM-ablated animals (*Figure 6D–F*). Overall, cooling induced reversals at a higher probability in RIM-ablated animals compared to the wild type. This difference was particularly notable for persistent periods of cooling lasting from 7 to 30 s (*Figure 6E*). We also examined the probability with which forward runs end after a period of cooling (*Figure 6—figure supplement 1C–E*). We found that, compared to the wild type, forward runs in RIM-ablated animals are more likely to end after a period of cooling regardless of run length (*Figure 6—figure supplement 1E*). Together, these analyses support a role of the RIM-dependent motor feedback in promoting a persistent forward run state.

## Agent-based simulations driven by a reduced model recapitulate the role of CD feedback in positive thermotaxis

To understand how CD might sustain motor states during thermotaxis, we built a minimal dynamical systems model of the thermotaxis circuit (*Figure 7*). In this model, temperature fluctuations encoded by a thermosensory neuron are conveyed to a downstream interneuron. The interneuron then outputs to a motor command neuron that determines the motor state. A copy of the motor command is then relayed back to the interneuron after being weighted by a feedback gain factor (*g*). A positive gain factor means the motor-related signal is reinforcing to the activity of the interneuron, effectively forming a positive feedback loop. A negative gain factor translates to a negative feedback loop. We allowed the gain factor to vary between 1 (positive feedback), 0 (no feedback), and −1 (negative feedback) to test the impact of the recurrent circuit motif on circuit output.

When exposed to oscillating inputs, the positive feedback model exhibited stable high and low states in both the interneuron and the motor neuron. The autocorrelative timescale of these states (a measure of persistence) outlasted the oscillatory period of the input signal (*Figure 7B*). This was not the case for the models with no feedback or negative feedback where the oscillatory signal remained evident in both the interneuron and the motor neuron (*Figure 7B*).

We then used this circuit model to simulate animal locomotion along linear thermal gradients (*Figure 7C*). The positive feedback model more effectively drove migration up the thermal gradient than the model with no feedback. In contrast, the negative feedback model exhibited less effective thermotaxis than the no feedback model (*Figure 7C, D*). The duration of forward runs was overall significantly longer and exhibited stronger dependence of run duration on run direction in the positive feedback model compared to the alternative models (*Figure 7E*). Lastly, forward runs were much more likely to terminate after a period of cooling in the no feedback model compared to the positive feedback model, consistent with experimental observations in RIM-ablated animals. Thus, the positive feedback model best explained the neural activity and behavioral data. These results

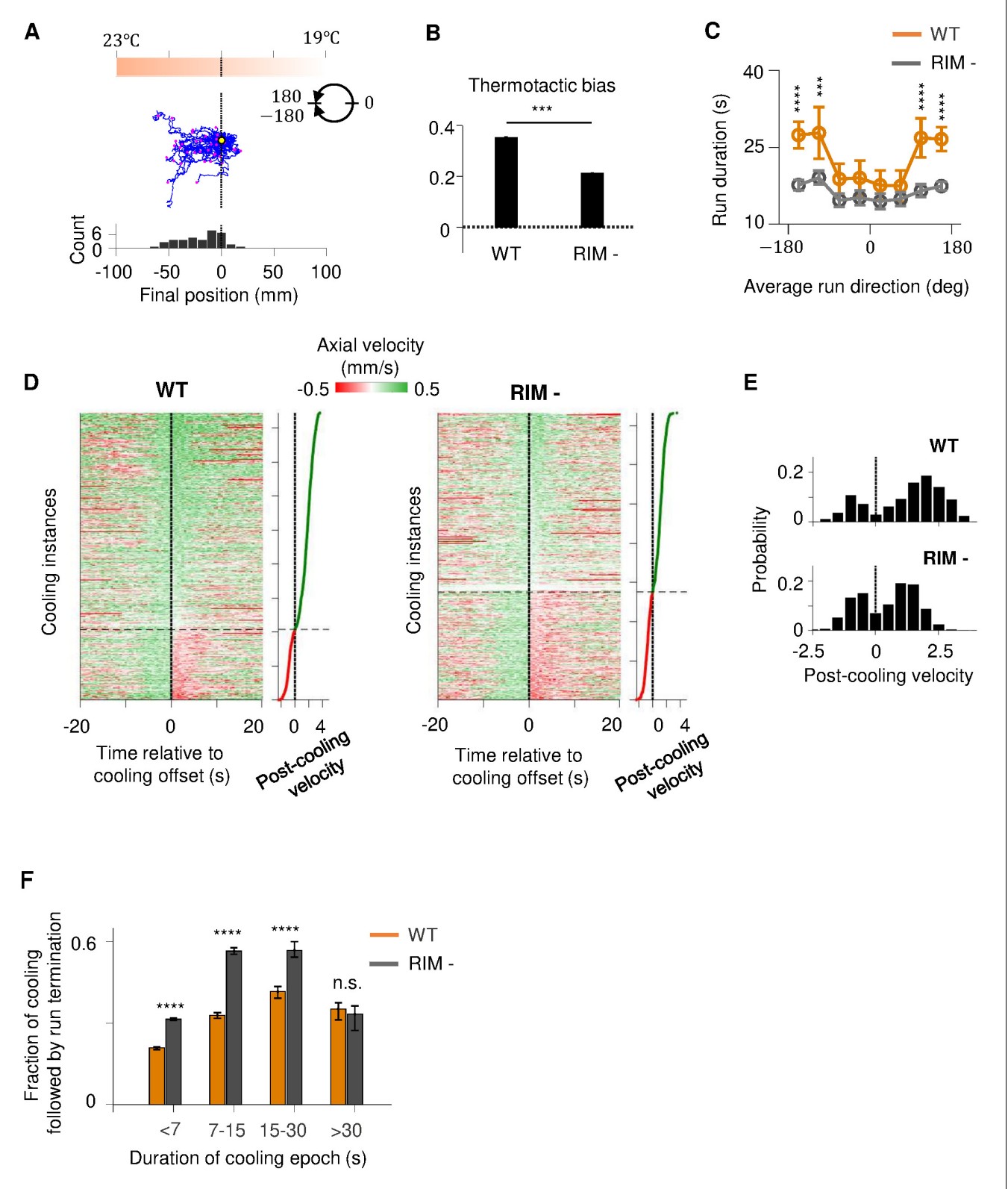

**Figure 6.** RIM ablation disrupts positive thermotaxis and leads to increased susceptibility to sensory fluctuations. (A) Example trajectories of RIM-ablated animals (N = 39) cultivated at 25°C and exposed to the same thermal gradient as in *Figure 1A*. Top: schematic of the thermal gradient. Middle: trajectories of individual animals during positive thermotaxis. The starting points of all trajectories are aligned (yellow dot) and the end points are marked by magenta dots. Bottom: a histogram of the final location of the animals at the end of the 20 min period. (B) Average thermotactic bias of

*Figure 6 continued on next page*

*Figure 6 continued*

wild-type (N = 140) versus RIM-ablated animals (N = 102). (**C**) Forward run duration as a function of forward run direction in RIM-ablated animals (blue) compared to the wild type (gray). Error bars are standard errors of the mean (s.e.m.). Wilcoxon rank-sum test was used to compare the run duration of wild-type versus RIM-ablated animals. p<*0.05, **0.01, ***0.001, p>0.05 (non-significant) for panels without asterisk. (**D**) Velocity profiles of wild-type (left) and RIM-ablated (right) animals aligned to the end of cooling epochs that occurred during forward runs. Heat maps are generated by concatenating velocity profiles from individual cooling epochs along the y-axis and sorting by the average velocity within the first 2 s after the offset of cooling epochs (shown as line plot to the right). Black dotted lines divide instances in which forward runs continued past the offset of cooling epochs from instances where reversals ensued within the first 2 min of cooling offset. (**E**) Histograms of post-cooling velocities in wild-type (top) and RIM-ablated animals (bottom). Analysis applied to same dataset as in (**D**). (**F**) Fraction of cooling epochs that were followed by transition from forward runs to reversals as a function of the duration of cooling epochs (orange: wild type, gray: RIM ablated).

The online version of this article includes the following source data and figure supplement(s) for figure 6:

**Source data 1.** Thermotaxis behavior in RIM ablated animals.

**Figure supplement 1.** RIM ablation results in higher likelihood of forward runs ending after a period of cooling.

---

indicate that CD in the form of positive feedback promotes persistent circuit activity states and effective navigation during time-varying thermosensory inputs.

## Discussion

We have uncovered a role for CD, a feedback signal from the motor circuit, in sustaining a neural state for forward locomotion during *C. elegans* thermotaxis. By relaying a copy of the motor command to a sensory processing interneuron, the thermotaxis circuit encodes a recurrent loop that integrates rapidly varying thermal inputs with more slowly varying motor state signals. This integration results in stable neural activity states that sustain the behavioral state corresponding to forward locomotion. Persistent neural activity states may enable the circuit to filter out rapid fluctuations in sensory input and promote efficient navigation in a dynamic sensory environment. During positive thermotaxis up gradients, *C. elegans* generates sustained periods of forward locomotion that carry it up temperature gradients; CD prevents these periods of forward locomotion from being curtailed by transient negative temperature fluctuations.

In mice and flies, recurrent circuitry has a prevalent role in persistent neural activities and behavioral states. In many systems, CD has a role in suppressing neural or behavioral responses. In contrast, we show that CD is also able to reinforce the behavioral response to a sensory input. In the *C. elegans* circuit for positive thermotaxis, the AIY interneuron receives the motor state signal. Because AIY is postsynaptic to many sensory neurons and is required for navigation in other modalities (*Wakabayashi et al., 2004*; *Tsalik and Hobert, 2003*; *Luo et al., 2014a*), CD-based feedback to AIY might play a general role in many different sensorimotor pathways.

The RIM interneuron is required for the propagation of the CD signal to AIY to promote positive thermotaxis. In a recent study of olfactory responses in immobilized animals, RIM was also shown to send feedback input to AIB, another sensory processing interneuron whose activity promotes the reversal state (*Gordus et al., 2015*). In the olfactory pathway, RIM activity was required to correlate the activity of AIB and the AVA premotor interneuron. Silencing RIM led to more reliable odor-evoked response in AIB and odor-induced initiation of forward runs. These findings are consistent with a model where a RIM-dependent CD couples AIB activity to the motor command signal, thereby preventing AIB and its downstream circuit from passively responding to fluctuating olfactory inputs. Thus, the effect of silencing RIM in the olfactory sensorimotor pathway mirrors the impact of RIM ablation on AIY, during positive thermotaxis, shown in this study. Together, this evidence suggests a wide role for CD in promoting persistent states in sensorimotor transformation.

Our findings add to a growing body of literature from across species that motor behavior can significantly impact sensory processing (*Petreanu et al., 2009*; *Zagha et al., 2013*; *Fu et al., 2014*; *Schneider et al., 2014*; *Seelig and Jayaraman, 2015*; *Ouellette et al., 2018*; *Musall et al., 2019*; *Stringer et al., 2019*; *Salkoff et al., 2020*). An explicit dependence of sensory encoding on behavioral states has been shown to contribute to variability in stimulus-evoked neural and behavioral responses (*Fontanini and Katz, 2008*; *McGinley et al., 2015*). In mice performing visual and auditory tasks, cortex-wide neural activity can be dominated by movement-related signals, many of which are uninstructed (*Musall et al., 2019*). Importantly, these movement-related signals closely

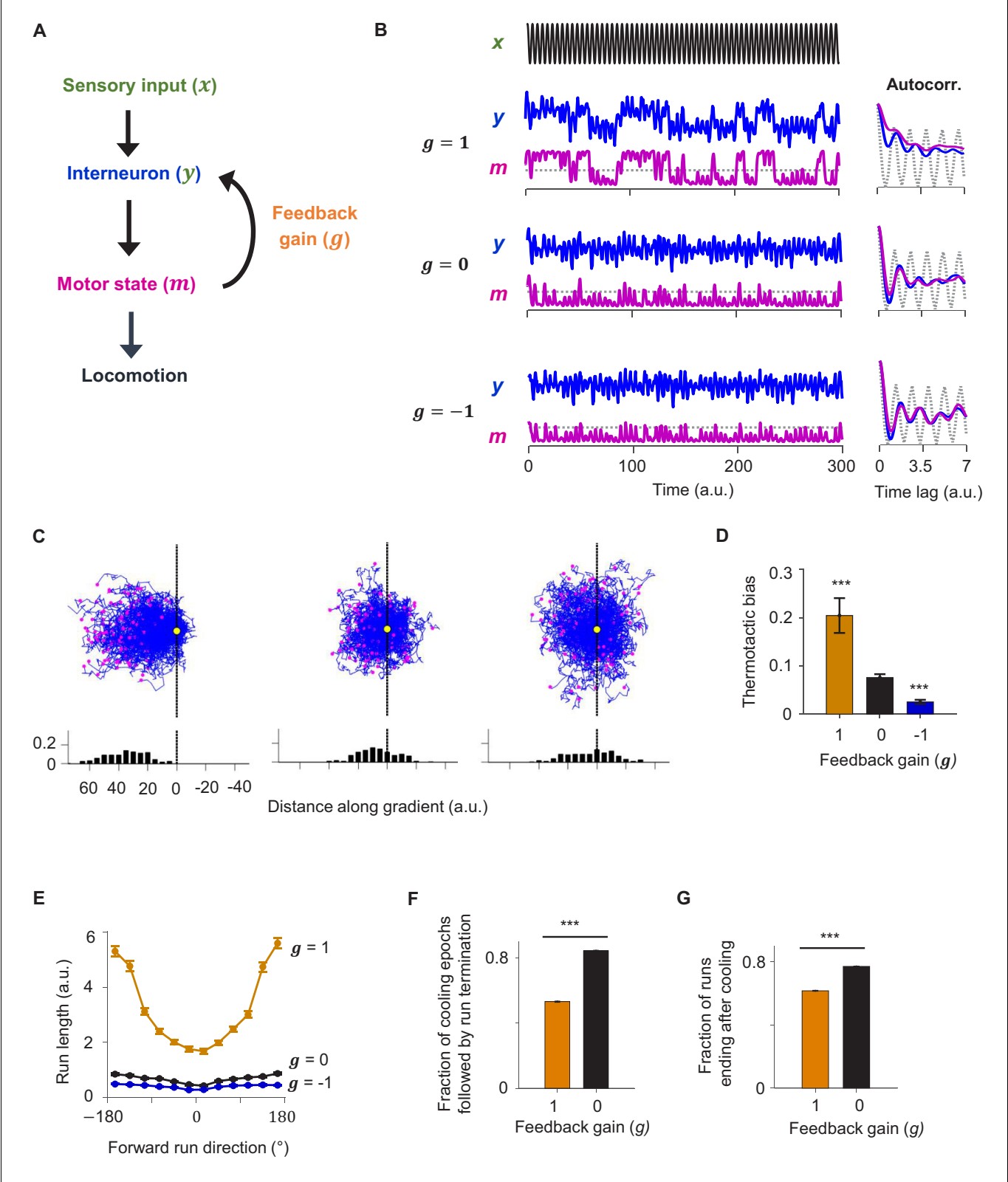

**Figure 7.** A reduced model explains the role of corollary discharge in sustaining forward locomotion during thermotaxis. (**A**) Schematic of the circuit model. (**B**) Dynamics of the model in response to an oscillating input stimulus. Top: temporal profile of the input signal. Middle: dynamics of the model with the feedback strength set to 1 (positive feedback), 0 (no feedback), or −1 (negative feedback). (**C**) Simulated trajectories of navigational behavior on a 2-D arena with linear input gradient, with feedback strength set to 1 (left), 0 (middle), and −1 (right). (**D**) Thermotactic biases for trajectories

*Figure 7 continued on next page*

*Figure 7 continued*

generated by models with feedback strength equaling 1 (positive feedback), 0 (no feedback), or −1 (negative feedback). (**E**) Forward run duration as a function of forward run direction for the behavioral simulations in (**D**). (**F**) Fraction of cooling epochs that were followed by the termination of forward runs in simulations with the feedback gain set to 1 (blue) or 0 (black). (**G**) Fraction of forward runs that ended after a period of cooling in simulations with the feedback gain set to 1 (blue) or 0 (black). Error bars are 95% CI. Wilcoxon rank-sum test *p<0.05; **p<0.01; ***p<0.001.

The online version of this article includes the following source data for figure 7:

**Source data 1.** Computational model of the thermotaxis circuit.

predict the inter-trial variability in neural response. In *C. elegans*, whole-brain imaging in both stationary and moving animals has shown brain-wide encoding of the forward run and reversal states (*Venkatachalam et al., 2016*; *Kato et al., 2015*; *Nguyen et al., 2016*). The roles of the AIY interneuron studied here and the AIB interneuron studied in *Gordus et al., 2015* may reflect a general use of the integration of sensory and motor-related signals during *C. elegans* navigation. In both cases, RIM is needed to propagate the motor-related signal to sensory processing interneurons.

In addition to its postsynaptic partners predicted by the connectome, RIM releases a diverse array of neuromodulatory signals (*Taylor et al., 2019*). Thus, through RIM and other neurons, the motor state signal may be broadcast to many neuron types, leading to correlated activity patterns throughout the *C. elegans* brain (*Kaplan et al., 2018*). This hypothesis may be tested with whole-brain imaging after RIM inactivation. We do not fully understand the synaptic mechanism by which CD reaches AIY. We observed that loss of tyramine, a biogenic amine produced by RIM, only partially disrupted the CD signal in AIY. Broadly disrupting biogenic amine synthesis or vesicle release from RIM also yielded similarly partial defects *Figure 4—figure supplement 2*.

The tyramine receptor, SER-2, is expressed in many neurons including AIY. Cell-specific perturbation or rescue of SER-2 function in AIY could help test the requirement of tyramine signaling in relaying the CD signal. Other signaling molecules are likely involved as well, and more extensive molecular and cellular dissection is needed to understand how the CD signal reaches AIY. Interestingly, a recent study demonstrated that hyperpolarization of RIM extended the duration of the forward runs during spontaneous locomotion through RIM-specific function of the gap junction protein UNC-9 (*Sordillo et al., 2021*). Exploring the cell-specific involvement of gap junction genes in relaying CD could also elucidate the molecular basis of feedback to AIY.

## Materials and methods

### Molecular biology and transgenic strain construction

#### Promoters

The following promoters were used to allow neuron-specific expression of a calcium sensor, chrimson, and miniSOG. Most were generated from genomic DNA isolated from mixed stage N2 animals. Promoters include 4.8 kb (P*rig-3*), 0.9 kb (P*inx-1*), 5.3 kb (P*glr-1*), 2.9 kb (P*cex-1*), 0.86 kb (P*lgc-55B*), and 3.1 kb (P*nmr-1*) genomic sequence. All promoters except P*nmr-1* and P*lgc-55B* used the genomic sequence of the respective length starting immediately upstream of the predicted ATG start codon of the respective genes. For P*nmr-1*, a 2 kb internal fragment that reduces the 5.1 kb *nmr-1* reporter expression was removed (*Kawano et al., 2011*). Details on P*lgc-55B* can be found in *Gao et al., 2015*. See *Appendix 1—table 1* for the full list of constructs and transgenes used in this study.

#### Calcium imaging

For AIY calcium imaging, *aeals003* was generated by integrating *olaEx1621* [P*mod-1::GCaMP6s*; P*ttx-3::RFP*; P*unc-122::mCherry*]. The integrant was outcrossed against N2 for four times to generate strain ADS003 and crossed into *lite-1* to generate QW1410. As in previous studies, temperature-evoked AIY activity was reliably recorded from neurites as opposed to the soma (*Clark et al., 2006*; *Biron et al., 2006*).

For AFD calcium imaging, *aeals004* was generated by integrating an existing Ex line [P*gcy-8::GCaMP6s*; P*gcy-8::RFP*; P*unc-122::mCherry*]. The integrant was outcrossed against N2 for four times to generate strain ADS004.

**Table 1.** Parameter values used in *Figure 7B*.

| Parameter | Value |
| --- | --- |
| $\tau_1$ | 2/3 |
| $\tau_2$ | 2/3 |
| $g_{L1}$ | 1 |
| $g_{L2}$ | 1 |
| $w_0$ | 1 |
| $w_{fb}$ | −1, 0, or 1 |
| $w_{21}$ | 1 |
| $k_1$ | 15 |
| $k_2$ | 5 |
| $k_3$ | 5 |
| $\beta_1$ | 0 |
| $\beta_2$ | 1.5 |
| $\beta_2'$ | 1.5 |
| $\gamma_1$ | 1 |
| $\gamma_2$ | 1 |
| $C_1$ | 0.5 |
| $C_2$ | 0 |

For premotor interneuron and motor neuron calcium imaging, pJH3338 was constructed for calcium imaging for premotor interneurons and head motor neurons. The GCaMP6s reporter was optimized for *C. elegans* and contained three *C. elegans* introns (*Lim et al., 2016*; *Chen et al., 2013*). GCaMP6s was fused with codon-optimized mCherry (wCherry) at the C-terminus to facilitate ratiometric measurement via simultaneous imaging of GFP and RFP. The reporter expression was driven by P*glr-1* as described above. This construct was co-injected with lin-15(+) marker to *lin-15(n765)* animals to generate extrachromosomal transgenic array *hpEx3550* and subsequently integrated to generate *hpIs471*. The integrated array was outcrossed against N2 wild type four times to generate ZM8558. For simultaneous AIY and premoter/interneuron imaging, *hpIs471* was crossed with *aeaIs003* to generate ADS027.

## Neuron ablation

pJH2829, pJH3311, pJH2931, pJH2890, and pJH2827 were constructed for LED-based neuronal ablation for RIM, AIB, AVA (plus other neurons), AVB (plus other neurons), and AVA/AVE/AVD/RIM/PVC (plus other neurons), respectively. miniSOG fused with an outer mitochondrial membrane tag TOMM20 (tomm20-miniSOG or mito-miniSOG) (*Qi et al., 2012*; *Shu et al., 2011*). An inter-cistronic sequence splice leader (SL2) was inserted between the coding sequence of tomm20-miniSOG and codon-optimized mCherry (wCherry; a gift of A Desai, UCSD) to visualize neurons that express miniSOG and to examine the efficacy of ablation. SL2 sequence was PCR amplified off the splice leader sequence (SL2) between *gpd-2* and *gpd-3*. These constructs were co-injected with the lin-15(+) marker in *lin-15(n765)* animals to generate extrachromosomal arrays *hpEx2997*, *hpEx3464*, *hpEx3072*, *hpEx3064*, and *hpEx2940*, respectively. With the exception of *hpEx3072*, other arrays were integrated to generate *hpIs327*, *hpIs465*, *hpIs331*, and *hpIs321*. All integrated transgenic arrays were outcrossed four times against N2, except *hpIs327*, which was outcrossed seven times against N2, before being used for behavioral analyses or to be combined with AIY calcium imaging analyses or behavioral analyses.

### AIY imaging upon neuronal ablation

*aeaIs003* was crossed with *hpIs327*, *hpIs321*, *hpEx3072*, *hpIs331*, and *hpIs465*, respectively, to generate ADS010, ADS014, ADS026, ADS036, and ADS046. They were used for AIY calcium imaging upon ablation of RIM, premotor interneurons (with a few other neurons), and AIB, respectively.

### AIY calcium imaging upon genetic manipulation of synaptic transmission and optogenetic stimulation

For AIY imaging in genetic synaptic transmission mutants, QW1408, QW1409, QW1411, QW1175, and QW1415 were generated by crossing *aeaIs003* into the corresponding mutant backgrounds listed in *Appendix 1—table 1*.

For AIY imaging upon cell-type-specific manipulation of synaptic transmission, *aeaIs003* was crossed with *yxIs25*, *xuEx1414*, and *kyEx4962* to generate ADS043, ADS042, and ADS013, respectively (*Li et al., 2014*; *Zhang et al., 2005*; *Gordus et al., 2015*).

Chrimson (*Klapoetke et al., 2014*) was codon-optimized and fused at C-terminus with wCherry as described (*Lim et al., 2016*). Chrimson expression was driven by P*lgc-55B* and P*rig-3* to generate *pHR2* and *pHR6*. These constructs were co-injected with P*ges-1::GFP* into QW1410 to generate *aeaEx003* (ADS29) and *aeaEx005* (ADS31) for AIY imaging upon optogenetic stimulation of AVB and AVA, respectively.

*aeaEx003* and *aeaEx005* were then crossed into *hpIs327;aeaIs003;lite-1* to generate ADS033 and ADS035 for AIY calcium imaging in RIM-ablated animals, upon AVB and AVA stimulation, respectively.

## Behavioral assays

### Positive thermotaxis assay

L4 animals were cultivated at 25°C the night before the assay. On the day of the experiment, the behavioral arena was allowed to equilibrate until a stable linear thermal gradient spanning 19 –23°C was established. Before each assay session, a thin layer of NGM agar sized 20 cm on each side was placed on the arena and allowed to equilibrate to the temperature of the arena. Twenty young adults were collected from their cultivation plates and briefly washed in NGM buffer before they were transferred onto the thin agar. These animals were allowed to explore the assay environment for 5 min before behavioral recording starts. Afterwards, a CMOS camera positioned above the arena recorded continuously every 500 ms for 20 min. Animal trajectories were extracted from the raw behavioral recordings using custom-written LABVIEW software. Subsequent analyses were performed in MATLAB.

### Spontaneous locomotion assay

Animals were cultivated and prepared for behavioral assay in identical manners as for the positive thermotaxis assay. The same behavioral arena, equilibrate to room temperature (22°C), was used to assay spontaneous locomotion. Behavioral recordings were conducted the same way as in the positive thermotaxis assay. Subsequent analyses were performed using the same LABVIEW software as above and subsequently in MATLAB.

### Calculation of thermotactic bias

For each animal, the instantaneous velocity ($v$) and speed ($v$) were calculated from the animal's centroid positions. The velocity vector was then projected onto direction of the thermal gradient, which in this case was parallel to the negative direction of the x-axis of the behavior arena. The thermotactic bias is the ratio between the velocity projection along the thermal gradient and the instantaneous speed of the animal:

$$\text{thermotactic bias} = \frac{-v_x}{v}$$

## Calcium imaging in semi-constrained animals

### Sample preparation and imaging setup

L4 larval animals expressing cytosolic GCaMP6s::wCherry were cultivated at 25°C the night before the imaging experiment. Immediately before the imaging session, animals were transferred to a microscope slide with a 5% agarose pad (2 mm thick). A small drop of NGM buffer was added to the agarose pad and a #1 coverslip was lowered onto the pad. This preparation allowed the animal enough mobility to execute head and (partial) body oscillations characteristic of forward runs and reversals. Under this preparation, the animal exhibits slow, local displacements, but cannot fully leave the field of view. Calcium imaging was performed on an upright spinning disc confocal microscope (Nikon Eclipse LV100 and Yokogawa CSU22) and iXon3 DU-897 EMCCD camera (Andor). High-resolution images were collected through a 40×, 0.95 NA Nikon Plan Apo lambda objective. 3D volumetric stacks were acquired in both the green (GCaMP6s) and red (wCherry) channels with an exposure of 30 ms at approximately 1.2 volumes per second.

### Control of thermal stimulation

Animals were imaged on a custom-built temperature control stage where a PID controller and H-bridge amplifier (Accuthermo) drove a thermoelectric cooler (TEC) (Newark) that pumped heat into and out of a thin copper plate with a liquid-cooled water block (Swiftech) acting as a thermal reservoir. A type-T thermocouple microprobe (Physitemp) was placed on the copper plate underneath a thin steel tab. A custom written Labview program was used to specify the desired temperature waveform.

### Extraction of calcium transient levels

To extract fluorescence intensities for individual neurons, we identified connected regions above a predefined intensity threshold and registered these regions of interest (ROI) across a movie based on spatial proximity across frames. For each ROI, we computed mean intensity of the top 30 pixels in both the green (GCaMP6s) and the red (wCherry) channel. The instantaneous activity of each neuron was computed using the following equation:

$$\Delta R(t)/R_0 = (R(t) - R_0)/R_0$$

$R$ is the ratio between the mean intensity value in the green channel and that of the red channel. $R_0$ represents the lowest 1st percentile of $R(t)$ values in the time series.

### Event-triggered averages (ETAs) of neural activity

AIY activity was extracted from a defined window of time spanning the event interest (i.e., onset of thermal stimuli, transition between the forward run and reversal states, onset and offset of AVA activation). For analysis in *Figure 3D*, AVA ON and OFF states were defined by thresholding AVA activity using the Otsu method, which was implemented in MATLAB using the multithresh function. Data across multiple epochs of the same event were concatenated into matrices and presented as heat maps throughout the paper. Average ETA profiles were computed by averaging data from that precede or lag the event of interest by the same number of time points.

### Duration of AIY activation

AIY activity data from each imaging session was fit using a Gaussian mixture models (see 'Statistical analysis' for implementation details). The number of Gaussian components was determined by iteratively fitting models with component number k = 1,2,3,...,6 and assessing the quality of the fit using the Bayesian information criterion (BIC). As k increases, the BIC value typically drops sharply from k = 2 to k = 3 and varies little after that. For wild-type data, models with k = 3 consistently capture the baseline AIY activities during reversals, the heightened activities during forward runs, and the large transients that occur at the onset of the forward runs or in response to warming stimuli. We thus defined the intersection point of the two Gaussian components with the lowest and the second lowest mean as the threshold above AIY is considered to be in the activated state. We then use this threshold value to binarize the AIY activity time series and compute the duration of each bout of AIY activation.

## Optogenetic stimulation and simultaneous calcium imaging

Experimental animals expressing Chrimson were grown on NGM plates supplied with 5 µM all-trans retinal (ATR) mixed with OP50 bacteria. Control animals of the same genotypes were grown on NGM plates seeded with OP50 without ATR. The day before the experiment L4 animals were picked onto fresh plates (with ATR for the experimental groups and without ATR for the control groups). On the day of the experiment, young adult animals were prepared for imaging in the semi-constrained preparation as described above. During imaging, pulses of red light were delivered from a filtered white LED lamp. Pulse timing was controlled by MATLAB scripts. For calcium imaging, animals were illuminated with only the blue laser (488 nm) to avoid strong activation of Chrimson.

## Neuron ablation

Transgenic animals expressing miniSOG were collected from late L1 to L2 stage onto a small NGM plate (3.5 cm diameter). The plate was placed under a blue LED spotlight (Mightex, peak wavelength 617 nm) for 40 min. Following illumination, the animals were allowed to recover for overnight at 15° C to examine the disappearance of cells. All ablation was performed using animals that carried integrated miniSOG transgens, with the exception for AVA ablation. Ablation of AVA was carried out in animals that carried an extrachromsomal array for *Prig-3-miniSOG-SL2-RFP*, which was subjected to random loss during somatic division. Animals used for ablation were selected for those that did not show expression (hence ablation) in a pharyngeal neuron that affects the survival of ablated animals.

## Statistical analysis

### Statistical tests

The Wilcoxon rank-sum test was used in the following comparisons: (1) comparing calcium activity upon the initiation of forward runs or reversals between wild-type animals and various neuron-ablation experiments, (2) comparing the probability of change in AIY activity upon the initiation of forward run or reversals between wild-type and AIY::TeTX animals, and (3) comparing the thermotactic bias between wild-type and RIM-ablated animals. To control for multiple comparison, p values were adjusted using the Benjamini–Hochberg correction. 95% confidence intervals were determined by bootstrapping.

### Gaussian mixture model

Gaussian mixture models were fit to AIY activity distributions from each independent dataset using the fitgmdist function in MATLAB. The initial cluster centers were generated through the k-means++ algorithm, and the initial mixing proportions were set to uniform.

## Modeling of circuit activity and behavior

### Neural circuit model

We use a minimal model to capture the interaction between the key components of the thermotaxis circuit:

$$\text{Thermosensory neuron (AFD)} - V_0: \quad \tau_0 \frac{dV_0}{dt} = -g_{L0}(V_0 - V_{L0}) + I_{input}R \tag{1}$$

$$\text{Interneuron (AIY)} - V_1: \quad \tau_1 \frac{dV_1}{dt} = -g_{L1}(V_1 - V_{L1}) + F_{10}V_0 + F_{12}V_2 \tag{2}$$

$$\text{Motor command neurons} - V_2: \quad \tau_2 \frac{dV_2}{dt} = -g_{L2}(V_2 - V_{L2}) + F_{21}V_1 \tag{3}$$

where $g_{L1}$, $g_{L2}$, and $g_{L3}$ are leak conductances and are non-negative. $V_{L1}$, $V_{L2}$, and $V_{L3}$ are the resting potentials. Synaptic interactions are modeled as linear or sigmoidal functions:

$$F_{10}(V_0) = w_0 V_0 \tag{4}$$

$$F_{12}(V_2) = w_{fb}\left(\frac{1}{1+e^{-k_1 \times (V_2 - \beta_1)}}\right) \tag{5}$$

$$F_{21}(V_1) = w_{21}\left(\frac{\gamma_1}{1+e^{-k_2 \times (V_1 - \beta_2)}}\right) - \left(\frac{\gamma_2}{1+e^{-k_3 \times (V_1 - \beta_2')}}\right) \tag{6}$$

where $w_0$, $w_{fb}$, and $w_{21}$ are the network weights, and $g_i$, $k_i$, and $\beta_i$ define the height, steepness, and inflection point of sigmoidal functions. The two terms in $F_{32}$ represent separate groups of premotor interneurons that promote forward runs (e.g., AVB) or reversals (e.g., AVA). *Equation 6* essentially performs a max operation between the two terms to determine whether the motor output favors forward runs ($F_{32}>0$) or reversals ($F_{32}<0$).

The above model is further simplified by setting AFD activity to its steady-state value, $V_0(t) \cong V_0 = V_{L0} + \frac{I_{input}R}{g_{L0}}$, which reduces the model to two dimensions:

$$\tau_1 \frac{dV_1}{dt} = -g_{L1}V_1 + \alpha I_{input} + F_{12}(V_2) + C_1 \tag{7}$$

$$\tau_2 \frac{dV_2}{dt} = -g_{L2}V_2 + + F_{21}(V_1) + C_2 \tag{8}$$

where $\alpha = \frac{w_0 R}{g_{L1}}$, $C_1 = g_{L1}V_{L1} + V_{L0}$, and $C_2 = g_{L2}V_{L2}$.

Based on dynamical systems theory (*Strogatz, 2015*), the above model has two distinct stable states as long as their nullclines intersect at least three times, which can be achieved by a wide range of parameter values. *Table 1* lists the parameter values that were used for network simulations in *Figure 7B*. Note that $w_{fb}$ is varied between −1, 0, and 1, corresponding to negative feedback, no feedback, and positive feedback networks.

## Simulation of thermotaxis behavior

We used the same network model described above to drive agent behavior. The locomotory state $M(t)$ of the animal is determined by the activity of the motor command neuron:

$$M(t) = 1, \text{ if } V_2>0 \rightarrow \text{agent executes forward run}$$

$$M(t) = 0, \text{ if } V_2<0 \rightarrow \text{agent executes reversal}$$

At the start of each forward run, the new heading direction is chosen randomly from a uniform distribution with range [-180°, 180°]. During an ongoing forward run or reversal, the heading direction, $\theta(t)$, was kept constant. When a forward run ends and a reversal state starts, the heading direction changes by 180°:

$$\theta(t) = \begin{cases} (\cos\theta_0, \sin\theta_0) & t=0 \\ \theta(t-dt) & M(t) = M(t-dt) \\ -\theta(t-dt) & M(t) = 1 \text{ and } M(t-dt) = -1 \\ (\cos\theta_0, \sin\theta_0) & M(t) = -1 \text{ and } M(t-dt) = 1 \end{cases} \tag{9}$$

These simple behavioral rules allows us to specifically model the biased random walk component of thermotaxis, while leaving the RIM-independent klinotaxis component out of the analysis.

All agents are simulated to move at constant speed (one unit length per time step) on a two-dimensional linear thermal gradient. The gradient is set to lie along the x-axis: $T(x) = c_T x$, where $x$ represents the x coordinate. Since AFD is known to sense temporal changes in temperature, the input current to AFD evoked by thermal stimuli is defined by $I_{input}(t) = c_T(x(t) - x(t - \Delta t))$, where $x(t)$ is the instantaneous x-position of the agent.

Each simulation is initialized by setting an starting position of (0,0), an initial heading angle drawn from the uniform distribution from [−180°, 180°], an initial network state of ($V_1(t_0) = 1$, $V_2(t_0) = 1$), and with the animal in a forward run state. Upon numerical integration, simulated worms move autonomously in their environment for a predetermined duration ($t_{max}$).

## Acknowledgements

We thank Daniel Witvliet for insights on *C. elegans* connectomes and generating *Figure 2B*. We thank the laboratories of Yun Zhang, Daniel Colón-Ramos, Shawn Xu, and Cori Bargmann for strains. This work was supported by NIH P01 GM103770 (ADTS), NIH R01 NS082525-01A1 (ADTS, MZ, MJA), NIH R01 GM084491 (MJA), the Burroughs Wellcome (VV), and the CIHR foundation 154274 (MZ). We thank Steven Flavell and members of the Samuel, Alkema, and Zhen laboratories for constructive advice and help in completion of the study and manuscript preparation.

## Additional information

### Funding

| Funder | Grant reference number | Author |
| --- | --- | --- |
| National Institute of Neurological Disorders and Stroke | NS082525-01A1 | Mark J Alkema<br>Mei Zhen<br>Aravinthan DT Samuel |
| National Institute of General Medical Sciences | PO1 GM103770 | Aravinthan DT Samuel |
| National Institute of General Medical Sciences | RO1 GM084491 | Mark J Alkema |
| Burroughs Wellcome Fund | | Vivek Venkatachalam |
| Canadian Institutes of Health Research | 154274 | Mei Zhen |

The funders had no role in study design, data collection and interpretation, or the decision to submit the work for publication.

### Author contributions

Ni Ji, Conceptualization, Resources, Methodology, Data curation, Software, Formal analysis, Validation, Investigation, Visualization, Writing - original and revised manuscript; Vivek Venkatachalam, Resources, Software, Methodology, Writing - original draft, Writing - review and editing; Hillary Denise Rodgers, Wesley Hung, Taizo Kawano, Christopher M Clark, Maria Lim, Resources; Mark J Alkema, Mei Zhen, Resources, Supervision, Writing - original draft, Project administration, Writing - review and editing; Aravinthan DT Samuel, Conceptualization, Supervision, Funding acquisition, Writing - original draft, Project administration, Writing - review and editing

### Author ORCIDs

Ni Ji https://orcid.org/0000-0002-7870-0678
Vivek Venkatachalam https://orcid.org/0000-0002-2414-7416
Hillary Denise Rodgers http://orcid.org/0000-0002-0565-1940
Mark J Alkema https://orcid.org/0000-0002-1311-5179
Mei Zhen https://orcid.org/0000-0003-0086-9622
Aravinthan DT Samuel https://orcid.org/0000-0002-1672-8720

### Decision letter and Author response

Decision letter https://doi.org/10.7554/eLife.68848.sa1
Author response https://doi.org/10.7554/eLife.68848.sa2

## Additional files

### Supplementary files

- Transparent reporting form

## Data availability

All data generated or analysed during this study are included in the manuscript and supporting files. Source data files have been provided for Figures 1-6. Source code has been provided for Figure 7.

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

# Appendix 1

**Appendix 1—table 1.** Constructs and transgenic arrays.

**Calcium imaging**

| Plasmid | Injection marker | Transgene | Strain |
|---|---|---|---|
| pDACR1286[*Pmod-1::GCaMP6s*] (25 ng/μl); pDACR63[*Pttx-3::mCherry* ] (25 ng/μl) | pDACR218[*Punc-122:: dsRed*] (40 ng/μl) | *aeaIs003(AIY) (integrated olaEx1621\*)* | ADS003 |
| pDACR943[*Pgcy-8::GCaMP6s*] (30 ng/μl); pDACR801 [*Pgcy-8::mCherry*] (5 ng/μl) | pDACR20[*Punc122::GFP*] (20 ng/μl) | *aeaIs004 (AFD) (integrated olaEx1527)\** | ADS004 |
| pJH3338[*Pglr-1-GCaMP6s::wCherry*] | pL15EK*lin-15AB* genomic DNA ( 20 ng/μl) | *hpIs471 (premotor/ motor)* | ZM8558 |

**Optogenetic stimulation**

| | | | |
|---|---|---|---|
| pHR2[*Plgc-55B-Chrimson::wCherry*] | pL15EK[*lin-15AB* genomic DNA] (80 ng/μl) | *aeaEx003 (AVB/others)* | ADS029 |
| pHR6[*Prig-3-Chrimson::wCherry*] | pL15EK[*lin-15AB* genomic DNA] (80 ng/μl) | *aeaEx005 (AVA/others)* | ADS031 |

**Cell ablation**

| | | | |
|---|---|---|---|
| pJH2829[*Pcex-1- MiniSOG::SL2::wCherry*] | pL15EK[*lin-15AB* genomic DNA] (20 ng/μl) | *hpIs327 (RIM)* | ZM7978 |
| pJH3311[*Pinx-1- MiniSOG::SL2::wCherry*] | pL15EK[*lin-15AB* genomic DNA] (20 ng/μl) | *hpIs465(AIB)* | ZM8484 |
| pJH2931[*Prig-3- MiniSOG::SL2::wCherry*] | pL15EK[*lin-15AB* genomic DNA] (20 ng/μl) | *hpEx3072 (AVA/others)* | ZM7198 |
| pJH2890[*Plgc-55B- MiniSOG::SL2::wCherry*] | pL15EK[*lin-15AB* genomic DNA] (20 ng/μl) | *hpIs331(AVB/others)* | ZM7297 |
| pJH2890[*Pnmr-1-MiniSOG::SL2::wCherry*] | pL15EK[*lin-15AB* genomic DNA] (20 ng/μl) | *hpIs321(AVA/E/D/RIM/ PVC/others)* | ZM7054 |

Synaptic manipulation

| | | | |
|---|---|---|---|
| *Pttx-3::TeTx::mCherry* **Zhang et al., 2005** | | *yxIs25 (AIY)* | ZC1952 |
| *Ptdc-1::TeTx::mCherry* **Gordus et al., 2015** | | *kyEx4962 (RIM/RIC)* | CX14993 |

\*Gift of Daniel Colon-Ramos.

# Appendix 2

**Appendix 2—table 1.** Strains.

**For thermotaxis and locomotion assays**

| Strain | Genotype | Purpose | Figure |
|---|---|---|---|
| Bristol N2 | Wild type | Wild-type behavior | *Figure 1A–D* |
| ZM7978 | *hpIs327* | Behavior upon RIM ablation | *Figure 6* |
| **Calcium imaging** | | | |
| ADS003 | *aeasIs003* | AIY imaging | *Figures 2E–I, 3A, B* |
| ADS004 | *aeaIs004* | AFD imaging | *Figure 2C, D* |
| ADS027 | *aeaIs003; hpIs471* | Simultaneous imaging of AIY, AVA, RME, SMDD, SMDV, and RIM | *Figure 2B, E* |
| ADS043 | *aeaIs003; yxIs25* | AIY imaging, upon blockade of AIY chemical transmission | *Figure 3B* |
| ADS010 | *aeaIs003; hpIs327* | AIY imaging, upon ablation of RIM | *Figures 4A, B, 5A, C* |
| ADS014 | *aeaIs003; hpIs321* | AIY imaging, upon ablation of RIM, AVA, AVE, AVD, and PVC | *Figure 4A* |
| ADS026 | *aeaIs003; hpEx3072* | AIY imaging upon ablation of AVA | *Figure 4A* |
| ADS036 | *aeaIs003; hpIs331* | AIY imaging, upon ablation of AVB | *Figure 4A* |
| ADS046 | *aeaIs003; hpIs465* | AIY imaging, upon ablation of AIB | *Figure 4A, B* |
| ADS029 | *aeaEx003; aeaIs003; lite-1(ce314)* | AIY imaging upon optogenetic stimulation of AVB | *Figure 4C* |
| ADS031 | *aeaEx005; aeaIs003; lite-1(ce314)* | AIY imaging upon optogenetic stimulation of AVA | *Figure 4E* |
| ADS033 | *aeaEx005; aeaIs003; hpIs327; lite-1(ce314)* | AIY imaging, upon RIM ablation and AVA stimulation | *Figure 4E* |
| ADS035 | *aeaEx003; aeaIs003; hpIs327; lite-1(ce314)* | AIY imaging, upon RIM ablation and AVB stimulation | *Figure 4C* |
| ADS013 | *aeaIs003; kyEx4962* | AIY imaging, upon disruption of RIM/RIC chemical transmission | *Figure 4—figure supplement 2* |
| ADS006 | *aeaIs003;tdc-1(n3419)* | AIY imaging in tyramine/octopamine synthesis mutant | *Figure 4—figure supplement 2* |
| QW1411 | *aeaIs003; eat-4(ky5)* | AIY imaging in glutamate mutant | *Figure 4—figure supplement 2* |
| QW1175 | *aeaIs003; unc-31(e928)* | AIY imaging in dense core vesicle release mutant | *Figure 4—figure supplement 2* |
| QW1408 | *aeaIs003; cat-1(e1111)* | AIY imaging in biogenic amine transporter mutant | *Figure 4—figure supplement 2* |

