## [Decision Letter]

**Acceptance summary:**

In this manuscript Ji and colleagues report that information processing in a primary sensory interneuron of *C. elegans* is modulated by behavioral state: the AIY neuron encodes external stimuli relayed from thermo-sensory neurons exclusively during forward crawling but not during backward crawling. This modulation depends on the reversal interneuron RIM. The authors report evidence that this interaction is a corollary discharge, i.e. motor command copy, to stabilize the forward crawling state in the presence of transient aversive thermal fluctuations; thereby, corollary discharge prevents erratic forward-backward switches supporting robust thermotaxis. Both sensory and behavioral modulation of AIY was shown in previous studies, but it remained puzzling how these different inputs are utilized simultaneously in the context of a behaving animal. Moreover, the function of why a primary sensory interneuron is so strongly modulated by behavior was unclear. Similar phenomena can be observed in other model organisms ranging from worms, flies and mice but the underlying mechanisms and functions are still unclear. The current manuscript sheds more light in these directions showing that sensory processing even at its earliest stages is instantaneously modulated by behavior to support robust navigation.

**Decision letter after peer review:**

[Editors’ note: the authors submitted for reconsideration following the decision after peer review. What follows is the decision letter after the first round of review.]

Thank you for submitting your work entitled "Corollary Discharge Promotes a Sustained Motor State in a Neural Circuit for Navigation" for consideration by *eLife*. Your article has been reviewed by 3 peer reviewers, one of whom is a member of our Board of Reviewing Editors, and the evaluation has been overseen by a Senior Editor. The following individual involved in review of your submission has agreed to reveal their identity: M Eugenia Chiappe (Reviewer #3).

Our decision has been reached after a detailed follow-up consultation between the reviewers. Based on these discussions and the individual reviews below, we are convinced that your manuscript is potentially very important and a strong candidate for publication in *eLife*. Therefore, we reject your manuscript now but encourage you to address the concerns and resubmit at a later stage. If you decide to re-submit the paper, please provide a point-by-point response to this letter and the reviewer's comments.

Please pay particular attention to some of the major concerns that came-up also in the consultation session, which we summarize here:

1. The reviewers are concerned about how unconstrained the animals are in your assays. Based on the data you provide, they conclude that the animals must have been recorded under almost immobile conditions. Since your major conclusions are contingent on activity recordings made in freely behaving animals this caveat needs to be addressed by (a) showing data demonstrating unconstrained movement, i.e. locomotion speed and /or posture kymographs and (b), eventually provide additional activity data obtained from sufficiently unconstrained animals. It might be sufficient to record from individual neurons, like AIY, as long as this permits more natural movement and confirms your major results.

2. The ethological model lacks sufficient support by your data and analyses. The alternative model of an efference copy from RIM simply inhibiting AIY during reversals seems very likely. However, you propose an unexplained sign-inversion in the interactions between RIM and AIY, and it is unclear to us how such an interaction can lead to a persistent feedback gain from the motor circuit to AIY, subsequently maintaining forward motor state. To address this, you could show more directly that in RIM(-) animals reversal initiations are tightly locked to sensory input. These data should be retrievable from your thermotaxis assays and AFD/AIY imaging data, including the suggested new experiments above.

3. The reviewers raise concerns that some experiments were not performed with sufficient repetitions and lack statistics, therefore require more validation.

4. In the discussion, we agreed that if the ethological model finds better support in a new manuscript, deciphering the molecular pathway in more detail suggested by reviewer #2, might be of less priority. However, alternatively you could tone down the major statements of corollary discharge function and focus your studies on a more complete molecular characterization of RIM-AIY interactions.

*Reviewer #1:*

In this manuscript Ji and colleagues report that a primary sensory interneuron, AIY, is modulated by behavioral state (forward versus reverse crawling), which depends on the reversal interneuron RIM. The authors suggest that this interaction is a corollary discharge, i.e. motor command copy, to stabilize the forward crawling state in the presence of transient aversive sensory fluctuations during thermotaxis. Both sensory and behavioral modulation of AIY was shown in previous studies, but it remained puzzling how these inputs converge onto AIY and which circuits are involved. Moreover, the function of why a primary sensory interneuron might be so strongly modulated by behavior was unclear. Similar phenomena were observed in other model organisms ranging from worms, flies and mice and hence currently draw a lot of attention in the neuroscience community since mechanisms and functions are still pretty unclear; therefore, this work is potentially very important, and I am very enthusiastic about the main findings. However, at the current stage the study is very preliminary and suffering in various experiments from insufficient repetitions, lack of proper analyses and statistics. While my comments below are extensive, they require merely more experimental repetitions and additional analyses. Addressing them is essential for publication, but I believe this can be done within reasonable time.

1. Figure 1: during positive thermotaxis it is not surprising that animals experience short negative dT/dt fluctuations while performing forward runs. This is indeed expected, and animals can easily deal with this, both considering the classical random biased walk strategy (since switch to reversal in this model is probabilistic, a fraction of episodes of negative dT/dt will not elicit terminating the forward state). To convince readers of this manuscript that there is indeed an interesting and surprising observation in this data, the authors should perform some additional analyses showing how episodes of continuous negative and positive dT/dt are distributed in their duration, e.g. in a 2D histogram. The authors then should help the reader not familiar with the literature and refer to the reports showing that thermosensory neurons are indeed sensitive in such an operating range.

Moreover, an analysis should be performed showing that episodes of negative dT/dt occur indeed throughout forward runs, not just at the beginning or at the end.

2. Inspecting the Ca++-imaging data in moving animals (e.g. Figure 2), I conclude that these animals must be crawling at extremely slow rates. Recent work showed that SMD neuronal activity is tightly locked to head bending phase (e.g. (Hendricks et al., 2012; Kaplan et al., 2019; Yeon et al., 2018)), which in this paper appears to be as slow as 2min (see Figure 2D, 5A). This is 1-2 orders of magnitude slower than freely crawling worms under standard laboratory conditions (e.g. compare to SMD imaging data in ref (Kaplan et al., 2019) or in semi-restrained animals (Hendricks et al., 2012)). Moreover, AVA and RIM Ca++ traces in this study exhibit minutes-lasting plateau states, which is characteristic for immobilized animals and typically not seen in freely crawling animals (see (Kato et al., 2015) for a comparison of these neurons between immobilized and freely crawling conditions). In conclusion, animals in this study must be nearly immobilized. I assume this is because animals have been placed between a coverglass and a 5% agarose pad, the latter having an unusual high concentration. To a non-expert *eLife* reader these caveats are quickly overseen.

2a. The authors should provide data about crawling speed in these experiments.

2b. The authors should provide an in-depth explanation why they chose such extreme conditions; perhaps to avoid movement artefacts given the slow 1Hz acquisition rate? Or is AIY modulation only detectable in these extreme conditions with strong and prolonged RIM activity phases?

2c. Were thermotaxis assays in Figure 1 performed under these conditions (coverglass, 5% agarose)? Are animals able to perform thermotaxis under these conditions?

3. While AFD activity traces in Figure 2A look convincing, additional analysis is required for AIY and other inter- motor-neurons to demonstrate sensory responses and gating by behavioral state. The peak in power spectrum alone does not suffice to conclude that AIY etc. activity is stimulus locked. The AIY activity trace in Figure 2—figure supplement 1 from immobilized unstimulated worms indicates interesting dynamics in AIY, that certainly would appear as a peak in a power spectrum.

3a. I suggest showing phase aligned averages of the neurons and a statistical test showing that their activity gets significantly entrained.

3b. How do power spectra and phase-averages differ in forward versus reverse?

3c. Why does AIY show a peak at 0.067 and not 0.033Hz, like AFD?

3d. An equivalent control dataset from non-stimulated animals is essential.

4. The finding that AIY is modulated by reversal neuron RIM in immobilized worms is crucial for the main conclusion of this paper, since it distinguished corollary discharge / efference copy from behavioral feedback. However, this statement is based only on a single observation shown in Figure 2—figure supplement 1 and lacks any quantification. Is AIY activity different in RIM high versus low states? This should be scrutinized via an analysis like it was done for reversal states in Figure 3; with multiple animal repetitions and appropriate statistics. In direct comparison to freely moving worms, this would be an important result deserving a spot in a main Figure. Without this extra data and analysis, the authors' hypothesis that AIY modulation is corollary discharge / efference copy versus behavioral feedback is not sufficiently supported.

5. I am wondering how reversal and forward states were annotated in AVA or AVB ablated animals respectively, which should lack these behaviors according to a large body of literature. If AVA and AVB were successfully ablated, the analysis in Figure 4B should not be possible. Also, ablation of RIM and AIB can affect behavioral state durations and switching frequencies. The authors should provide these data along with evidence that the cells were indeed specifically ablated, thus they must validate and characterize all of their ablation strains. Data like in Figure 4C,D should be shown as well form the other ablations.

6a. Data shown in Figure 4E come from partially very low numbers of independent experiments and no statistics are provided. These shortcomings must be addressed.

6b. Moreover, stimuli were applied at very different pre-stimulus AIY activity baselines. More data would definitely strengthen this result substantially.

7. I find the result that RIM and SMD neurons show stimulus evoked activity fluctuations in RIM(-) animals very interesting, but this result needs more substantial quantifications (see comment (3)).

8. Figure 6: the effect of RIM ablation on thermotaxis is very interesting but there are many ways how this manipulation could affect the behavior. Additional analysis of the data could provide more compelling evidence supporting their model: consistent with their model run durations are decreased (Figure 6E). If the function of RIM corollary discharge is indeed to maintain prolonged forward states in spite of negative dT/dt epochs, run terminations should be coinciding with these events, but not in control animals. See also comment (1) for analyses suggestions.

9. Modulation of primary sensory circuits by behavioral state seems to extend to many other neurons in the worm brain (Kato et al., 2015), and this seems to be a more general phenomenon seen in other organisms (e.g. Aimon et al., 2019; Musall et al., 2019; Salkoff et al., 2019; Stringer et al., 2019). See also for a recent discussion (Kaplan et al., 2018). The current manuscript, in my opinion, is pretty important providing data hinting at a function of these phenomena. This is just my recommendation, but I think the authors could make more impact with their story by discussing their findings in the context of these recent discoveries.

*Reviewer #2:*

Ni Ji and colleagues apply quantitative analysis of behavior and mutli-neuronal calcium imaging to identify and examine a corollary discharge circuit that filters sensory input based on motor state during thermotaxis. The main claims of the paper are that RIM reports this motor state (the corollary discharge signal) to the first layer interneuron AIY, which acts as a sensory filter to ignore temperature fluctuations encoded by AFD during reversals:

1. AIY encodes temperature and motor state information

2. Motor state representations in AIY arise from a corollary discharge signal originating from RIM.

3. CD allows positive feedback to sustain forward locomotion

The first and second claims are well-supported, while the ethological model is less well-developed and was not rigorously tested. Issues with the data, particularly what seem like unrealistically long reversal states make some results difficult to interpret.

Overall, I am convinced by the data showing that RIM broadcasts a motor state signal (the CD) that filters or modulates upstream sensory pathways. This is a nice result but given the systemic effects of RIM/tyramine on sensory neurons and interneurons and the prior results showing that RIM ablation affects reversal rates even under isotropic conditions, I don't think the argument that it is supporting efficient thermotaxis is very well supported. Identifying the relevant receptor on AIY would bring better resolution to the circuit analysis.

1. The behavior and activity patterns look unusual for locomoting animals. 2A shows reversals that last 30-60 seconds, typical reversals during navigation are <5s. I looked at the dataset and indeed the F/R ethogram in 2A is incorrect-there are many more state changes than represented in the figure. However, the state durations for reversals are still very long (median >30s for reversals, just under 30s for forward). It does not seem like this can be correct for a freely moving animal and is inconsistent with the tracks shown in 1A.

2. As the authors note, SMDD/V activity has been reported to match head movements (Hendricks et al., 2012; Yeon, Kim et al., 2018; Kaplan, Thula et al., 2019), and so in panel 2D I would expect oscillatory activity that matches the forward gait in the traces and appears at ~1Hz in the power spectrum. As in 2A (and throughout), the extremely long "reversal" state durations look more typical of restrained animals.

3. Figure 2A and 2D. Do the colors in the activity histogram correspond to forward and reverse states? If so this should be indicated and perhaps match the color used for the ethogram. I am not sure having the Ca++ traces superimposed on the stimulus indicator and ethogram improves clarity, but it's fine.

4. The power spectra are too cramped and low-res in Figure 2D to be legible. It would be useful to compare spectra (or perform temp-AIY cross correlations) from stimulus cycles that occur during reversals vs forward epochs, particularly to support the claim about AIY.

5. How many animals are represented in the summary data (histograms/spectra) in Figure 2? Statistical tests on summary data should be reported.

6. Because AIY shows discrete Ca++ peaks, a quantification of warming cycles that do and do not produce an AIY response (as a binary output) in forward and reverse states might be more clear or intuitive than the (also small and cramped) histogram.

7. There is a formatting error in the legend of 2A (fstim, stim should be all subscript)

8. Label forward and reverse states in panel 2D as in 2A.

9. What do the colors mean in 2B?

10. I think the standard wiring diagram does not have an EJ between AFD and AIZ, as shown in 2B and 4A (though these authors may know better)?

11. Figure 2—figure supplement 1. Same questions as above regarding the very long reversals and lack of SMDD activity matching forward gait. The "right" side of panel A is labeled panel B and panel B as described in the legend and line 141 (neuronal cross correlations with RIM) is missing.

12. Figure 3. Consider using a perceptually uniform color map for heat maps (here and elsewhere).

13. Spell out corollary discharge in the Figure 3 legend.

14. While AVA correlates perfectly with reversals, it is less clear how well RIM does under different conditions (Gray et al., 2005; Guo et al., 2009; Piggott et al., 2011, Gordus et al., 2015)..… AVA activation/reversals can happen while RIM is inactive, and RIM activation can occur in the absence of a reversal, and in fact can lower the probability of a stimulus-evoked reversal (Gordus), while ablating RIM can increase reversal frequency (Gray, Piggott). While RIM Ca++ does correlate pretty well with reversal, and it's clear here that it is inducing sensory filtering at AIY, it's possible that it is not exclusively or necessarily reversals. This should be considered.

15. LGC-55 and TYRA-2 are expressed in AIB and AIZ; SER-2 and TYRA-2 are expressed in AIY (https://cengen.shinyapps.io/SCeNGEA/). Given that SER-2 mediates inhibition of synaptic transmission in GABAergic motor neurons (as some of these authors showed), it seems a promising receptor to mediate RIM's inhibitory effect. Was this checked? Identifying the AIY receptor would resolve some points below.

16. The principle claim of the paper is that the CD signal from RIM communicates a motor state to AIY to filter AFD input during reversals. If the effect is in large part tyramine-mediated, there are receptors for this modulator on many or most of the sensory neurons upstream of AIY, as well as on its interneuron synaptic partners. Are AFD responses normal during manipulation of RIM? Could other sensory neurons have altered responses to temperature that impact AIY? It would be show that a tyramine receptor acts in AIY to directly respond to RIM signaling (probably SER-2) and show that expression in AIY is necessary/sufficient to filter AFD signaling during reversals.

17. Should indicate color coding for WT and RIM- in 6E/F in the panel.

18. Previous results showing that ablating RIM can increase reversal frequency under some conditions, including an isotropic environment (Gray, Piggott), complicate the interpretation here that shorter run lengths in RIM-ablated animals under thermal fluctuation are due to the sensory filtering observed in AIY. Do RIM- animals reverse more under these conditions in the absence of thermal stimulation? While the need to maintain forward movement during fluctuations is clear, it is less clear why an animal (or AIY at least) would need to ignore AFD input during reversals. I'm particularly concerned about this aspect given the unusually long reversal states observed here.

Reviewer #3:

Here, Ji and colleagues study a sensory-driven navigational circuit and ask how sustained locomotor-related signals arise from self-generated, rapid changes in sensory stimuli driven by the navigational strategy of the worm to successfully reach its goal. The question on the relation among sensory cues, internally generated signals, and motor strategies, is poorly understood for any moving animal. Therefore, I consider the work very timely and important for a wide neuroscience audience.

I think this work is important and heroic: examining the presence and role of internal signals is no easy duty, and I think the authors are taking advantage of the model system beautifully. The propose role for a CD signal presented here is a very interesting and exciting idea, and consistent with the effect on behavior on the ablation of RIM (shown to be the source of motor-related signals of the AIY interneuron). However, the authors present some statements that I am not sure are shown directly from their experimental data. The concern is really about the gap between the description of the activity in AIY, a convincing metric of the presence of a persistent motor activity, and the interpretations of the data by the authors. While RIM ablation induces bleed through of sensory signals in motor cells, this effect may be due to the connections between RIM and motor cells per se rather than a consequence of the presence of a positive feedback CD signal at AIY level.

Through a phenomenological model, which was inspired by their experiments (combining analysis of behavior, physiology, and activity manipulations), they describe how the presence of a positive feedback CD signal can account for a persistent motor-related signal at an interneuron of the circuit, and the biased behavior of the worm. While this is a very exciting idea, I am not convinced that the experimental evidence supports this view clearly.

1. The authors show the presence of an internally generated signal at AIY reporting motor state, and convincingly show that RIM activity is required for this signal.

2. However, the increase of AIY activity at onset of Forward runs, and it decrease at onset of Reversals can be equally explained by either the a conventional efference copy (EC)-like signal that would be associated with reversals (since AIY is part of the forward pathway) or the proposed positive feedback CD signal (PF-CD) that would be associated with forward runs, and induce persistent activity in AIY. I see no effort in the manuscript to distinguish between these two possibilities. An EC like signal can also explained the increase in sensory cues in the response profile of AIY under oscillating temperature cues in the absence of activity in RIM (Figure 6A).

3. Optogenetic experiments are more consistent with an inhibition from reversal pathways (EC-like phenomena) than an excitation signal from forward runs (PF-CD like phenomena) (Figure 4). figure 4E, I really cannot see the effect the authors describe for the manipulation of AVB (either in the presence or absence of a functional RIM), whereas the description of the effect of the manipulation of AVA is closer to what we see in the data.

4. There is a missing metric for persistent activity in AIY, or a clear explanation of whether bimodality would be such metric. If this is the case, the authors make a very weak argument on how such a metric would measure RIM-dependent persistent motor-sate activity in AIY. If I understood correctly, bimodality should represent a more persistent activity at a particular motor state. But it is not very clear how bimodal the AIY activity is under the presence vs the absence of RIM activity, for example, if one compares figure 5A with Figure 2 Suppl Figure 1 (left column, mobile animals at constant T). Therefore, perhaps it is important to show the corresponding distribution for the supplementary figure to be able to compare the critical metric that is then quantified in Figure 5B.

5. Perhaps they could use the model to predict the effect on behavior of an EF vs PF_CD function. While the latter was examined, it is unclear how the worm behavior departs from a model under an EC signal at AIY

6. I suggest the authors make less of strong statements when the experimental evidence is not shown directly (for example, in line 235, they describe that "motor-related signals and Temperature-related signals reinforce each other": this is only shown through the model).

7. An exciting observation is the bleed through of sensory signals to motor cells within the circuit, and what potential impact these may convey at the level of the behavior of the worm (Figure 6). AIY has increased sensory components, but the lack of an EC could also induce such an effect, for example, by extending the time the neuron responds to T fluctuations.

8. What is the impact of making AIY insensitive to the motor state on other postsynaptic elements of the circuit? The authors show that motor cells themselves respond to sensory fluctuations likely affecting the persistent behavioral state. But is this a consequence of the lack of motor state sensitivity of AIY or a consequence of the lack of activity of RIM, a neuron that is presynaptic to motor cells? In conveying the idea of a PF-CD signal, it is important to distinguish these scenarios. A neuron that integrates information from AIY and AFD, and that is presynaptic to motor cells may be an ideal candidate. For example, AIB is very well suited for this distinction. If AIY activity contained both a persistent signature of motor state and thermal signals, then AIB sensory information should be modulated by the AIY's indirect input. On the other hand, if the motor state modulation in AIY is related to an EC signal, AIB cell sensory activity should not be strongly modulated by the lack of such EC signal (given electrical coupling of AIB and AFD). If this is not a possible experiment, then the strength of the statements should be adjusted accordingly, and the discussion on the presence of a conventional EC signal should be present.

---

## [Author Response]

[Editors’ note: the authors resubmitted a revised version of the paper for consideration. What follows is the authors’ response to the first round of review.]

Reviewer #1:In this manuscript Ji and colleagues report that a primary sensory interneuron, AIY, is modulated by behavioral state (forward versus reverse crawling), which depends on the reversal interneuron RIM. The authors suggest that this interaction is a corollary discharge, i.e. motor command copy, to stabilize the forward crawling state in the presence of transient aversive sensory fluctuations during thermotaxis. Both sensory and behavioral modulation of AIY was shown in previous studies, but it remained puzzling how these inputs converge onto AIY and which circuits are involved. Moreover, the function of why a primary sensory interneuron might be so strongly modulated by behavior was unclear. Similar phenomena were observed in other model organisms ranging from worms, flies and mice and hence currently draw a lot of attention in the neuroscience community since mechanisms and functions are still pretty unclear; therefore, this work is potentially very important, and I am very enthusiastic about the main findings. However, at the current stage the study is very preliminary and suffering in various experiments from insufficient repetitions, lack of proper analyses and statistics. While my comments below are extensive, they require merely more experimental repetitions and additional analyses. Addressing them is essential for publication, but I believe this can be done within reasonable time.

We thank the reviewer for recognizing the significance of our study. As suggested, we have substantially augmented our datasets and added analyses that strengthen our conclusions.

1. Figure 1: during positive thermotaxis it is not surprising that animals experience short negative dT/dt fluctuations while performing forward runs. This is indeed expected and animals can easily deal with this, both considering the classical random biased walk strategy (since switch to reversal in this model is probabilistic, a fraction of episodes of negative dT/dt will not elicit terminating the forward state). To convince readers of this manuscript that there is indeed an interesting and surprising observation in this data, the authors should perform some additional analyses showing how episodes of continuous negative and positive dT/dt are distributed in their duration, e.g. in a 2D histogram. The authors then should help the reader not familiar with the literature and refer to the reports showing that thermosensory neurons are indeed sensitive in such an operating range.Moreover, an analysis should be performed showing that episodes of negative dT/dt occur indeed throughout forward runs, not just at the beginning or at the end.

We agree. In any biased random walk, an animal will experience thermal fluctuations. With the minimal rules of a biased random walk, an animal would be able to migrate in the correct direction. We assert that a biased random walk can work *better* with an additional feature. If runs that happen to be pointed in the right direction are insulated from thermal fluctuations, they would not end prematurely and navigation would improve. This is borne out by numerical simulation and phenomenological modeling.

As requested, we have more extensively analyzed the trajectories of animals performing thermotaxis. These data indicate that the worm does suppress its sensitivity to thermal fluctuations, enabling the animal to sustain runs that carry it in the correct direction.

As now shown in Figure 1-Supplement 1, over half of forward runs exhibited by wild-type animals involved at least one cooling episode with a minimum drop in temperature exceeding 0.001 ^◦^C that did not occur at the beginning or the end of the run (Figure 1-Supplement 1A). Most of these episodes were shorter than 2-3 s (Figure 1-Supplement 1B) and involved temperature drops greater than 0.01 ^◦^C (Figure 1-Supplement 1C). Previous work has shown that *C. elegans* reliably responds to temperature change as small as 0.005 ^◦^C/s^1^. Negative thermal fluctuations in the middle of forward runs up temperature gradients are common and within the detectable range for *C. elegans*. This raises the question: how is the response to these fluctuations suppressed so that the run continues?

In our revision, we clarify that the behavior we investigate is a biased random walk with an augmentation. We found that animals are more likely to have persistent runs during forward movements in favorable directions. Because the principal thermosensory neuron in *C. elegans* is exquisitively sensitive to rapid temperature changes, there must be a neural mechanism that filters rapidly fluctuating thermal input during the forward motor state.

We found that the mechanism for suppressing thermal fluctuations involves feedback from the motor circuit to a first-layer interneuron. This feedback, in effect, turns persistent forward movements during positive thermotaxis into an attractor-like behavioral state that is less sensitive to transient negative thermal fluctuations.

The revised manuscript clarifies our contribution and conclusion. Thank you for the advice that led to these improvements.

2. Inspecting the Ca++-imaging data in moving animals (e.g. Figure 2), I conclude that these animals must be crawling at extremely slow rates. Recent work showed that SMD neuronal activity is tightly locked to head bending phase (e.g. (Hendricks et al., 2012; Kaplan et al., 2019; Yeon et al., 2018)), which in this paper appears to be as slow as 2min (see Figure 2D, 5A). This is 1-2 orders of magnitude slower than freely crawling worms under standard laboratory conditions (e.g. compare to SMD imaging data in ref (Kaplan et al., 2019) or in semi-restrained animals (Hendricks et al., 2012)). Moreover, AVA and RIM Ca++ traces in this study exhibit minutes-lasting plateau states, which is characteristic for immobilized animals and typically not seen in freely crawling animals (see (Kato et al., 2015) for a comparison of these neurons between immobilized and freely crawling conditions). In conclusion, animals in this study must be nearly immobilized. I assume this is because animals have been placed between a coverglass and a 5% agarose pad, the latter having an unusual high concentration. To a non-expert eLife reader these caveats are quickly overseen.2a. The authors should provide data about crawling speed in these experiments.2b. The authors should provide an in-depth explanation why they chose such extreme conditions; perhaps to avoid movement artefacts given the slow 1Hz acquisition rate? Or is AIY modulation only detectable in these extreme conditions with strong and prolonged RIM activity phases?

Reviewer 1 is correct. In experiments where the animal was moving very slowly, they were semi-constrained. The head region moved forward or backward without leaving the field of view at high magnification. Under high mechanical load, animals exhibited much slower head oscillations.

We used slowly moving animals during calcium imaging to optimize signal-to-noise and reduce movement artifact. This is particularly crucial for the AIY interneuron where sensory encoding is only detectable in its neurite and not its soma. This requires imaging at considerably higher resolution than other methods we have used^2^.

We more explicitly describe the semi-constrained preparation. We have also added new analyses that demonstrate how we infer motor states in semi-constrained animals during calcium imaging (Figure 2-Supplement 1). With our high-resolution of a portion of the head, we cannot use the axial speed of the animal that we and others use at lower magnification. Here, we segment motor states using the phase difference in the movement of two body landmarks (the AIY soma versus neurite). We confirm that this metric works when segmenting motor states in freely moving animals (Figure 2-Supplement 1A,B), providing us with a means of assessing forward vs backward movement in our calcium imaging experiments with semi-constrained animals (Figure 2-Supplement 1D,E). In both cases, the calcium dynamics in AIY neurite is positively correlated with axial (forward) velocity (Figure 2-Supplement 1C,F).

In the revision, we more clearly describe how we infer the forward or backward state based on the relative movement of internal markers, not by directly tracking crawling speed. Given that the undulation is roughly 0.1Hz, as the Reviewer rightly points out, we can infer that the crawling speed in calcium imaging experiments is roughly ten-fold lower than that of typical behavioral assays.

2c. Were thermotaxis assays in Figure 1 performed under these conditions (coverglass, 5% agarose)? Are animals able to perform thermotaxis under these conditions?

All behavioral assays were performed on large (20 cm X 20 cm), unseeded, 2% agarose plates with no coverslips placed on top. In the absence of bacterial food and the larger plate size, conditions used for assaying thermotaxis behavior are identical to those used for cultivating the animals.

3. While AFD activity traces in Figure 2A look convincing, additional analysis is required for AIY and other inter- motor-neurons to demonstrate sensory responses and gating by behavioral state. The peak in power spectrum alone does not suffice to conclude that AIY etc. activity is stimulus locked. The AIY activity trace in Figure 2—figure supplement 1 from immobilized unstimulated worms indicates interesting dynamics in AIY, that certainly would appear as a peak in a power spectrum.3a. I suggest showing phase aligned averages of the neurons and a statistical test showing that their activity gets significantly entrained.3b. How do power spectra and phase-averages differ in forward versus reverse?3c. Why does AIY show a peak at 0.067 and not 0.033Hz, like AFD?3d. An equivalent control dataset from non-stimulated animals is essential.

We agree. The power spectrum is not the best measure of stimulus-locked activity in AIY in response to thermal oscillations. The power spectrum may be affected by motor-related oscillations from locomotion and is less reliable for analyzing signals in short time windows, e.g. a short bout of forward or backward movement.

Throughout the revision, we now use the more direct measure of cross-correlation between neural activity and the temperature waveform. We compare this cross-correlation between the forward and reverse motor state. This improved analysis is easier to understand and confirms our previous conclusion, as shown in new Figures 2, 5. We conclude that AIY does not exhibit stimulus-locked response to temperature up-sweeps or down-sweeps in wild type animals, while it does exhibit phase-locked response to the thermal inputs in RIM-ablated animals.

4. The finding that AIY is modulated by reversal neuron RIM in immobilized worms is crucial for the main conclusion of this paper, since it distinguished corollary discharge / efference copy from behavioral feedback. However, this statement is based only on a single observation shown in Figure 2—figure supplement 1 and lacks any quantification. Is AIY activity different in RIM high versus low states? This should be scrutinized via an analysis like it was done for reversal states in Figure 3; with multiple animal repetitions and appropriate statistics. In direct comparison to freely moving worms, this would be an important result deserving a spot in a main Figure. Without this extra data and analysis, the authors' hypothesis that AIY modulation is corollary discharge / efference copy versus behavioral feedback is not sufficiently supported.

We have now analyzed AIY activity profile across multiple immobilized animals. To test whether AIY still carry motor-related signals, we aligned AIY activity to the onset and offset of AVA activation. In Figure 3C-D, we show that AIY activity rises around the activation and dips around the inactivation of the premotor interneuron AVA (note that the activation of AVA, defined by binarizing AVA activity using the Otsu method, may not be a perfect predictor of the onset of fictive reversal states). In addition, we present the average cross-correlations of AIY activity with other neurons in immobilized animals in Figure 3-Supplement 2B,C, which shows that the anti-correlation between the activity of AIY and that of AVA remains unchanged in immobilized animals.

5. I am wondering how reversal and forward states were annotated in AVA or AVB ablated animals respectively, which should lack these behaviors according to a large body of literature. If AVA and AVB were successfully ablated, the analysis in Figure 4B should not be possible. Also, ablation of RIM and AIB can affect behavioral state durations and switching frequencies. The authors should provide these data along with evidence that the cells were indeed specifically ablated, thus they must validate and characterize all of their ablation strains. Data like in Figure 4C,D should be shown as well form the other ablations.

It is not true that ablation of single premotor interneurons such as AVA or AVB completely abolishes reversals or forward movement. These premotor interneurons are active during and contribute to these motor states, but animals lacking AVA can still crawl backward and animals lacking AVB can still crawl forward as noted in the first study to ablate these neurons^3^ and subsequent studies of the contribution of these premotor interneurons to locomotion^4^. Animals without premotor interneurons can still move, but less well.

In Figure 4B and Figure 4 Supplement 1A-C, we show the detailed behavioral-state-triggered AIY activity profiles for every ablation strain used in this study.

In our ablation experiments, we used the flavoprotein miniSOG to induce neuronal death by photoactivation^5^. To verify death, we expressed miniSOG in neurons along with cytoplasmic RFP. To ablate neurons, the transgenic strain was exposed to LED light as early larva and animals were examined in adults. We have verified the process of miniSOG-induced neuronal ablation using RFP markers^6^. Upon LED exposure, neurons die and disintegrate. Red debris is gradually cleared by other tissues, most prominently by muscles. Complete clearance takes approximately two days, by which time the larva have become adults. The clearance of neuronal RFP signals in adults is the criterion for successful ablation of the neuron, and is not attributable to photobleaching.

6a. Data shown in Figure 4E come from partially very low numbers of independent experiments and no statistics are provided. These shortcomings must be addressed. (6b) Moreover, stimuli were applied at very different pre-stimulus AIY activity baselines. More data would definitely strengthen this result substantially.

We have substantially augmented our optogenetic and control datasets and analyses.

In the new Figure 4C-F, we present both the raw heat maps and analyses quantifying the red-light-evoked response in AIY. The new data allow us to differentiate the effect of optogenetic stimulation at different pre-stimulus activity levels of AIY. We show that optogenetic activation of AVB induced a strong increase in AIY activity when AIY exhibited low pre-stimulus activity levels. Optogenetic activation of AVB evoked weak to no increase when AIY exhibited medium to high pre-stimulus activity levels. These data suggest that AIY cannot be activated further if its activity is already near its maximum.

For AVA optogenetic activation, we observed a rapid drop in AIY activity when AIY had medium to high pre-stimulus activity levels.

In contrast, optogenetic activation of AVA or AVB in RIM-ablated animals yielded no significant response in AIY. These new data substantiate the crucial requirement for RIM for AVBand AVA-activation evoked responses in AIY. These results point to a critical role for RIM in relaying signals from pre-motor interneurons to AIY.

7. I find the result that RIM and SMD neurons show stimulus evoked activity fluctuations in RIM(-) animals very interesting, but this result needs more substantial quantifications (see comment (3)).

We have added these quantifications in the state-specific correlation between the stimulus and activity for all neurons that we studied in RIM-ablated animals Figure 5A. These analyses reveal significant correlations between the thermal stimuli and the activity of AVA, SMDD, and SMDV.

We agree with the reviewer that these correlations that emerge specifically in RIM- animals is interesting. This observation is consistent with our model. RIM-mediated motor feedback, by driving persistent neural activity states, helps to insulate motor neurons from the effects of transient or fluctuating sensory inputs.

8. Figure 6: the effect of RIM ablation on thermotaxis is very interesting but there are many ways how this manipulation could affect the behavior. Additional analysis of the data could provide more compelling evidence supporting their model: consistent with their model run durations are decreased (Figure 6E). If the function of RIM corollary discharge is indeed to maintain prolonged forward states in spite of negative dT/dt epochs, run terminations should be coinciding with these events, but not in control animals. See also comment (1) for analyses suggestions.

We agree. In the new Figure 6D-F we have more carefully and thoroughly quantified the effect of cooling (*dT/dt <* 0) on run termination. In wild type animals, cooling epochs that occurred during a forward run led to run termination (i.e. switching from positive to negative velocity) about 25% of the time. In RIM-ablated animals, this likelihood increases to 37%.

We found that the probability of negative *dT/dt* leading to run termination depends on the duration of the cooling epoch. In both wild type and RIM-ablated animals, cooling periods longer than 7 seconds were more likely to be followed by the termination of the run. For longer cooling epochs, the increase in RIM-ablated animals from the wild-type is more striking: nearly 2-fold for epochs between 7-15 seconds (Figure 6F). We observed no change in the frequency of run termination for sustained cooling epochs (*>* 30*s*) after RIM ablation. RIM plays a role in sustaining forward movements for brief but not sustained cooling epochs.

In a complementary analysis, we examined the probability of forward runs being preceded by a period of cooling (Figure 6-Supplement 1C-E). We found that this probability increased from about 29% in wild type animals to 47% in RIM-ablated animals. Consistent with our model, cooling epochs are more likely to truncate forward runs in RIM-ablated animals.

Taken together, our new analyses confirm that RIM ablation increases the susceptibility of forward runs to be terminated by negative thermal fluctuations. Our interpretation is that the forward run is an attractor-like state that is mediated by RIM-dependent motor feedback. This feedback reduces the sensitivity of forward movements to transient thermal fluctuations or noise. This is described in the revision.

9. Modulation of primary sensory circuits by behavioral state seems to extend to many other neurons in the worm brain (Kato et al., 2015), and this seems to be a more general phenomenon seen in other organisms (e.g. Aimon et al., 2019; Musall et al., 2019; Salk_off_ et al., 2019; Stringer et al., 2019). See also for a recent discussion (Kaplan et al., 2018). The current manuscript, in my opinion, is pretty important providing data hinting at a function of these phenomena. This is just my recommendation, but I think the authors could make more impact with their story by discussing their findings in the context of these recent discoveries.

We agree. A growing body of literature across species report the encoding of behavioral states in sensory areas. Studies in flies and the electric fish have revealed a role for motor feedback in canceling sensory inputs caused by self-motion. Our study has uncovered another functional role for motor feedback, generating attractor-like persistent behavioral states that filter transient sensory fluctuations. We now reference these pertinent studies that provide context for our work.

Reviewer #2:Ni Ji and colleagues apply quantitative analysis of behavior and mutli-neuronal calcium imaging to identify and examine a corollary discharge circuit that filters sensory input based on motor state during thermotaxis. The main claims of the paper are that RIM reports this motor state (the corollary discharge signal) to the first layer interneuron AIY, which acts as a sensory filter to ignore temperature fluctuations encoded by AFD during reversals:1. AIY encodes temperature and motor state information2. Motor state representations in AIY arise from a corollary discharge signal originating from RIM.3. CD allows positive feedback to sustain forward locomotionThe first and second claims are well-supported, while the ethological model is less well-developed and was not rigorously tested. Issues with the data, particularly what seem like unrealistically long reversal states make some results difficult to interpret.Overall, I am convinced by the data showing that RIM broadcasts a motor state signal (the CD) that filters or modulates upstream sensory pathways. This is a nice result but given the systemic effects of RIM/tyramine on sensory neurons and interneurons and the prior results showing that RIM ablation affects reversal rates even under isotropic conditions, I don't think the argument that it is supporting efficient thermotaxis is very well supported. Identifying the relevant receptor on AIY would bring better resolution to the circuit analysis.

We appreciate the positive assessment and suggestions for improvement.

1. The behavior and activity patterns look unusual for locomoting animals. 2A shows reversals that last 30-60 seconds, typical reversals during navigation are <5s. I looked at the dataset and indeed the F/R ethogram in 2A is incorrect-there are many more state changes than represented in the figure. However, the state durations for reversals are still very long (median >30s for reversals, just under 30s for forward). It does not seem like this can be correct for a freely moving animal and is inconsistent with the tracks shown in 1A.2. As the authors note, SMDD/V activity has been reported to match head movements (Hendricks et al., 2012; Yeon, Kim et al., 2018; Kaplan, Thula et al., 2019), and so in panel 2D I would expect oscillatory activity that matches the forward gait in the traces and appears at ~1Hz in the power spectrum. As in 2A (and throughout), the extremely long "reversal" state durations look more typical of restrained animals.

Reviewer One raised the same comments, which we have addressed with new experiments and analyses. See above.

3. Figure 2A and 2D. Do the colors in the activity histogram correspond to forward and reverse states? If so this should be indicated and perhaps match the color used for the ethogram. I am not sure having the Ca++ traces superimposed on the stimulus indicator and ethogram improves clarity, but it's fine.

We agree. We have now modified the colors of the histograms to match those used in the ethogram.

4. The power spectra are too cramped and low-res in Figure 2D to be legible. It would be useful to compare spectra (or perform temp-AIY cross correlations) from stimulus cycles that occur during reversals vs forward epochs, particularly to support the claim about AIY.

We agree. As described in our response to Reviewer One, we have replaced the power spectra analyses with the better cross-correlation correlation analyses between stimulus-triggered calcium activity for AIY and other neurons with the thermal stimuli during forward movement and reversals (new Figure 2D and Figure 5A). Thank you for the suggestion. These analyses more clearly represent the phase-locked response of AIY to thermal inputs that occurs in RIM-ablated animals.

5. How many animals are represented in the summary data (histograms/spectra) in Figure 2? Statistical tests on summary data should be reported.

The activity histogram in Figure 2 and Figure 5 are shown only for the sample dataset to the left. The cross-correlograms and the newly added stimulus-triggered averages, however, are generated from data across multiple animals. We have now included the number of animals corresponding to these analyses in the figure captions.

6. Because AIY shows discrete Ca++ peaks, a quantification of warming cycles that do and do not produce an AIY response (as a binary output) in forward and reverse states might be more clear or intuitive than the (also small and cramped) histogram.

We agree. We now provide much more detailed information about the AIY response to warming and cooling in Figure 2D.

7. There is a formatting error in the legend of 2A (fstim, stim should be all subscript)

This power spectrum has been supplanted by cross-correlation analyses.

8. Label forward and reverse states in panel 2D as in 2A.

We have incorporated these state labels.

9. What do the colors mean in 2B?

The colors only highlight the neurons of key interest.

10. I think the standard wiring diagram does not have an EJ between AFD and AIZ, as shown in 2B and 4A (though these authors may know better)?

This is correct, the original connectome does not report gap junctions between AFD and AIZ, but we have observed gap junctions in all of connectomes that we recently reconstructed across the developmental time course (see http://www.nemanode.org)^7^.

11. Figure 2—figure supplement 1. Same questions as above regarding the very long reversals and lack of SMDD activity matching forward gait. The "right" side of panel A is labeled panel B and panel B as described in the legend and line 141 (neuronal cross correlations with RIM) is missing.

The moving worms in this analysis are semi-constrained, giving rise to the very long reversals. The figure legends in the original submission had an error. We now provide neuronal cross-correlations for all multineuron imaging datasets in the resubmission (see Figure 2 and Figure 3-Supplement 2).

12. Figure 3. Consider using a perceptually uniform color map for heat maps (here and elsewhere).

In the revised figures, the heat maps for temperature change, locomotion state and calcium activity is used consistently throughout. We wish to point out that, in Figure 2E and 5A, the warming periods might appear reddish as result of a visual illusion. The actual colors corresponding to warming are shades of yellow as indicated by the blue-yellow heat map.

13. Spell out corollary discharge in the Figure 3 legend.

Done.

14. While AVA correlates perfectly with reversals, it is less clear how well RIM does under different conditions (Gray et al., 2005; Guo et al., 2009; Piggott et al., 2011, Gordus et al. 2015)..… AVA activation/reversals can happen while RIM is inactive, and RIM activation can occur in the absence of a reversal, and in fact can lower the probability of a stimulus-evoked reversal (Gordus), while ablating RIM can increase reversal frequency (Gray, Piggott). While RIM Ca++ does correlate pretty well with reversal, and it's clear here that it is inducing sensory filtering at AIY, it's possible that it is not exclusively or necessarily reversals. This should be considered.

RIM has been studied in many different contexts, and we cite a number of these previous studies. In our hands, we have never observed an increase of RIM activity that was not correlated with reversals during spontaneous movement. RIM calcium dynamics may differ in other experimental paradigms. This is now discussed in the revised Discussion.

15. LGC-55 and TYRA-2 are expressed in AIB and AIZ; SER-2 and TYRA-2 are expressed in AIY (https://cengen.shinyapps.io/SCeNGEA/). Given that SER-2 mediates inhibition of synaptic transmission in GABAergic motor neurons (as some of these authors showed), it seems a promising receptor to mediate RIM's inhibitory effect. Was this checked? Identifying the AIY receptor would resolve some points below.16. The principle claim of the paper is that the CD signal from RIM communicates a motor state to AIY to filter AFD input during reversals. If the effect is in large part tyramine-mediated, there are receptors for this modulator on many or most of the sensory neurons upstream of AIY, as well as on its interneuron synaptic partners. Are AFD responses normal during manipulation of RIM? Could other sensory neurons have altered responses to temperature that impact AIY? It would be show that a tyramine receptor acts in AIY to directly respond to RIM signaling (probably SER-2) and show that expression in AIY is necessary/sufficient to filter AFD signaling during reversals.

We thank the reviewer these suggestions. We think that they offer excellent starting point to reveal the molecular and signaling underlying of this feedback mechanism. However we don’t expect these pathways are linear enough to be solved by one or two single mutant analyses. Many neurons express the same receptors (e.g. SER-2), and one single neuron expresses many receptors (e.g. SER-2 and TYRA-3). Previously studies have revealed extensive redundancy in these pathways. We have incorporated these comments in the discussion for our future studies in the revised manuscript.

17. Should indicate color coding for WT and RIM- in 6E/F in the panel.18. Previous results showing that ablating RIM can increase reversal frequency under some conditions, including an isotropic environment (Gray, Piggott), complicate the interpretation here that shorter run lengths in RIM-ablated animals under thermal fluctuation are due to the sensory filtering observed in AIY. Do RIM- animals reverse more under these conditions in the absence of thermal stimulation? While the need to maintain forward movement during fluctuations is clear, it is less clear why an animal (or AIY at least) would need to ignore AFD input during reversals. I'm particularly concerned about this aspect given the unusually long reversal states observed here.

There have been conflicting reports on the effect on manipulating RIM. In their independent studies of *C. elegans* locomotion, Kawano and Zhen (unpublished results) reached these conclusions: during spontaneous movements, activation of RIM is always associated with reversals, anatomic ablation of RIM does not alter spontaneous motor states; constitutive silencing of RIM with Kir2.1 or TWK-18 does not alter spontaneous motor states. The results reported in this study are consistent with their independent earlier assessments. In our discussion, we now point out that there have been conflicting reports, and some of this conflict may be due to differences in the experimental paradigm.

Reviewer #3:Here, Ji and colleagues study a sensory-driven navigational circuit and ask how sustained locomotor-related signals arise from self-generated, rapid changes in sensory stimuli driven by the navigational strategy of the worm to successfully reach its goal. The question on the relation among sensory cues, internally generated signals, and motor strategies, is poorly understood for any moving animal. Therefore, I consider the work very timely and important for a wide neuroscience audience.I think this work is important and heroic: examining the presence and role of internal signals is no easy duty, and I think the authors are taking advantage of the model system beautifully. The propose role for a CD signal presented here is a very interesting and exciting idea, and consistent with the effect on behavior on the ablation of RIM (shown to be the source of motor-related signals of the AIY interneuron). However, the authors present some statements that I am not sure are shown directly from their experimental data. The concern is really about the gap between the description of the activity in AIY, a convincing metric of the presence of a persistent motor activity, and the interpretations of the data by the authors. While RIM ablation induces bleed through of sensory signals in motor cells, this effect may be due to the connections between RIM and motor cells per se rather than a consequence of the presence of a positive feedback CD signal at AIY level.Through a phenomenological model, which was inspired by their experiments (combining analysis of behavior, physiology, and activity manipulations), they describe how the presence of a positive feedback CD signal can account for a persistent motor-related signal at an interneuron of the circuit, and the biased behavior of the worm. While this is a very exciting idea, I am not convinced that the experimental evidence supports this view clearly.1. The authors show the presence of an internally generated signal at AIY reporting motor state, and convincingly show that RIM activity is required for this signal.

We thank Reviewer 3 for the positive comments.

2. However, the increase of AIY activity at onset of Forward runs, and it decrease at onset of Reversals can be equally explained by either the a conventional efference copy (EC)-like signal that would be associated with reversals (since AIY is part of the forward pathway) or the proposed positive feedback CD signal (PF-CD) that would be associated with forward runs, and induce persistent activity in AIY. I see no effort in the manuscript to distinguish between these two possibilities. An EC like signal can also explained the increase in sensory cues in the response profile of AIY under oscillating temperature cues in the absence of activity in RIM (Figure 6A).

We agree with the reviewer that both phenomena can in principle arise from either negative feedback (NFB) or positive feedback (PFB) from the motor circuit. This ambiguity arises in part from the inherent limit of calcium imaging, which reports neural activity as a change in fluorescence relative to a baseline.

We thus re-examined our results for evidence that could differentiate between the NFB and the PFB model. We found two sets of evidence supporting the latter. First, reversal-associated inhibition to AIY, as part of the NFB-EC model, should not affect how AIY responds to thermal stimuli during the forward run. Instead, we found that the thermosensory response in AIY became more reliable during both forward runs and reversals upon RIM ablation (Figure 5D). This observation argues against a pure NFB model where AIY is simply inhibited during reversals. Through a dynamical systems model (Figure 7), we show that our results are better explained by a positive feedback network, which exhibits bi-stable dynamics that are resilient to rapid input fluctuations under both the ON (i.e. forward run) and OFF (i.e. reversal) states.

The second piece of evidence comes from examining the amplitude of AIY response to thermal stimuli. We focus specifically on warming-evoked response in AIY during forward runs, which should not be impacted by any reversal-associated inhibitory EC signal. As shown in the new Figure 5D, we observe in the wild type a long-tailed distribution, reflecting the occasional large increases in AIY activity upon warming. These large activity transients were abolished in RIM-ablated animals. This observation cannot be explained by a reversal-associated NFB model, as these responses occur during forward runs when AIY is already in its high state. A plausible explanation, from a dynamical systems perspective, is that there is an imaginary eigencomponent associated with the high state of AIY activity^8^, which could allow AIY activity to transiently leave the stable attractor state and reach even higher levels.

Together, this evidence supports the requirement of positive feedback in sustaining AIY activity during forward runs. Our results, however, do not exclude the existence of an NFB signal. As mentioned in the new Discussion, a recent study from Cori Bargmanns group has shown that RIM drives reversals through its chemical output, while stabilizing forward runs through its gap junction outputs ^9^. Thus, it is possible that the RIM-dependent motor feedback found in our study also has more than one function. We have incorporated the above analysis and discussion points in the revised manuscript, and we thank Reviewer 3 again for prompting us to think more deeply about our circuit model.

On a related note, we have observed that both classical and recent literature use the terms efference copy and corollary discharge as largely synonymous concepts. As we describe in the new Introduction, von Holst and Mittelstaedt proposed the term efference copy and R.W. Sperry coined corollary discharge in 1950 to describe similar phenomena in flies and fish. Both terms referred to an internal copy of the motor signal that functions to cancel sensory inputs evoked by self-motion. While subsequent studies in fish and flies tend towards the original usages of EC or CD in their respective fields, studies in other species use EC and CD interchangeably when referring to inhibitory motor feedback signals. As we clearly describe in the introduction, we follow the general usage of the term corollary discharge and note that we consider it a synonym for ”efference copy”.

3. Optogenetic experiments are more consistent with an inhibition from reversal pathways (EC-like phenomena) than an excitation signal from forward runs (PF-CD like phenomena) (Figure 4). figure 4E, I really cannot see the effect the authors describe for the manipulation of AVB (either in the presence or absence of a functional RIM), whereas the description of the effect of the manipulation of AVA is closer to what we see in the data.

We apologize for this mistake in our original submission. The heat map showing the AIY response to AVB stimulation was accidentally replaced with the heat map for the “no ATR control for AVA stimulation”. We have corrected this mistake in the new Figure 4C-F.

In addition, we have included more trials for each experimental condition. In Figure 4D,F, we quantify the stimulus-evoked response in AIY as a function of its pre-stimulus activity levels. We show that optogenetic activation of AVB induced a strong increase in AIY activity at low pre-stimulus activity levels, moderate increase for medium pre-stimulus activity, and no response at high pre-stimulus activity. This dependence likely reflects a ceiling effect, where AIY activity is not driven above a maximal value. We observe the opposite effect for the optogenetic stimulation of AVA. AIY exhibited a drop in activity upon AVA stimulation when AIY had medium to high pre-stimulus activity levels. RIM ablation shows that RIM is required for both AVA- and AVB-evoked changes in AIY activity.

These new data are consistent with our model that motor-state activity in AIY represents a feedback signal from premotor interneurons, and that this feedback is RIM-dependent.

4. There is a missing metric for persistent activity in AIY, or a clear explanation of whether bimodality would be such metric. If this is the case, the authors make a very weak argument on how such a metric would measure RIM-dependent persistent motor-sate activity in AIY. If I understood correctly, bimodality should represent a more persistent activity at a particular motor state. But it is not very clear how bimodal the AIY activity is under the presence vs the absence of RIM activity, for example, if one compares figure 5A with Figure 2 Suppl Figure 1 (left column, mobile animals at constant T). Therefore, perhaps it is important to show the corresponding distribution for the supplementary figure to be able to compare the critical metric that is then quantified in Figure 5B.

We thank the reviewer for this reasonable suggestion. In the new manuscript, we examined in closer detail the distribution of AIY in WT and RIM-ablated animals. As shown in Figure 3 Supplemental 1 and Figure 5 Supplemental 1, we found that in both WT and RIM-ablated animals, the distributions of AIY activity can be well approximated by a Gaussian mixture model consisting of three Gaussian distributions. The three Gaussian distributions can be viewed as representing low, intermediate and high activity states. In WT animals, the low state is most stable (represented by a high, narrow peak close to zero) while the high state (arising from large transients often observed at run onset) is broadly distributed. If one takes the low state as the OFF state and takes the sum of the intermediate and high states as the ON state, then the AIY ON state coincides closely with the forward run state in wild type animals Figure 3 Supplemental 1C.

In RIM ablated animals, the distribution of AIY activity is also best described by a mixture of three Gaussians (Figure 5 Supplemental 1C). Compared to the WT, the Gaussian associated with the low activity state is broader and the Gaussian corresponding to high activity represents a much smaller fraction of the full distribution. These changes reflect a less stable low state and the disappearance of large activity transients. If we again take the low state as the OFF state and takes the sum of the intermediate and high states as the ON state, we find that the duration of the AIY ON states were significantly shortened in RIM ablated animals under oscillating temperature (Figure 5D). This pattern is consistent with our earlier finding that forward run durations were shortened in RIM-ablated animals when exposed to varying temperature (Figure 6C and Figure 6 Supplement 1B). Interestingly, we did not find the duration of AIY activity to change significantly in RIM ablated animals under constant temperature, nor did the duration of forward runs. It is possible that other mechanisms are involved in regulating forward run durations in the absence of food and strong sensory inputs. In fact, *C. elegans* are known to gradually transition from frequent short forward runs (i.e. the local search state) to persistent forward runs (i.e. the global search state) within the first 20 minutes after transfer to an off-food environment without sensory gradients^10^.

Together, these results suggest RIM sustains AIY activity as well as prolongs the forward state in the presence of fluctuating thermal inputs. These new analyses on AIY distribution and duration of activation replace our previous analysis using an index of bimodality.

5. Perhaps they could use the model to predict the effect on behavior of an EF vs PF_CD function. While the latter was examined, it is unclear how the worm behavior departs from a model under an EC signal at AIY

We thank the reviewer for this suggestion. The goal of our model was indeed to explore the functional properties of different circuit architectures. As shown in the new Figure 7, we drew comparisons between networks models with the positive feedback (PFB), no feedback, or negative feedback (NFB) from the motor command neurons. We show that: (1) under oscillating sensory input, only the PFB model exhibited sustained ON and OFF states, as observed for AIY in WT animals (Figure 7B) in an agent-based simulation of thermotaxis, the PFB model exhibited stronger dependence of forward run duration on heading direction (i.e. biased random walk) and more efficient thermotaxis overall (Figure 7C-E). Furthermore, comparing the simulated trajectories generated through the PFB model and the model without motor feedback, we observe a stronger association between cooling stimuli and termination of forward runs. This observation echoes the difference in thermotaxis behavior between WT and RIM ablated animal shown in Figure 6D-F and Figure 6-S1C-E.

6. I suggest the authors make less of strong statements when the experimental evidence is not shown directly (for example, in line 235, they describe that "motor-related signals and Temperature-related signals reinforce each other": this is only shown through the model).

We have modified our statements to stay consistent with the data.

7. An exciting observation is the bleed through of sensory signals to motor cells within the circuit, and what potential impact these may convey at the level of the behavior of the worm (Figure 6). AIY has increased sensory components, but the lack of an EC could also induce such an effect, for example, by extending the time the neuron responds to T fluctuations.

As shown in the new Figure 5D, we found that the thermal response in AIY become more reliable during both forward runs and reversals. While a negative feedback EC signal could explain the increase in thermal response during reversals in RIM ablated animals, it could not explain the more reliable response during forward runs. As we demonstrate in the computational model (Figure 7B), a positive feedback network can generate bi-stable states, thereby preventing individual neurons from responding to rapid input fluctuations during both states.

8. What is the impact of making AIY insensitive to the motor state on other postsynaptic elements of the circuit? The authors show that motor cells themselves respond to sensory fluctuations likely affecting the persistent behavioral state. But is this a consequence of the lack of motor state sensitivity of AIY or a consequence of the lack of activity of RIM, a neuron that is presynaptic to motor cells? In conveying the idea of a PF-CD signal, it is important to distinguish these scenarios. A neuron that integrates information from AIY and AFD, and that is presynaptic to motor cells may be an ideal candidate. For example, AIB is very well suited for this distinction. If AIY activity contained both a persistent signature of motor state and thermal signals, then AIB sensory information should be modulated by the AIY's indirect input. On the other hand, if the motor state modulation in AIY is related to an EC signal, AIB cell sensory activity should not be strongly modulated by the lack of such EC signal (given electrical coupling of AIB and AFD). If this is not a possible experiment, then the strength of the statements should be adjusted accordingly, and the discussion on the presence of a conventional EC signal should be present.

Indeed, RIM is densely connected with many inter- and motor neurons and could affect behavior in different ways. Since AFD is the only thermosensory neuron known to contribute to positive thermotaxis^11^ and the only sensory neuron shown to phase-lock with thermal stimuli^12^, the stimulus-locked activity that emerges in the motor circuit after RIM ablation is likely downstream of AFD.

AIY and AIB are the two primary post-synaptic partners of AFD. A recent study showed that AIB receives excitatory feedback input from RIM and exhibits positively correlated activity with both RIM and the premotor neuron AVA^13^. Silencing RIM restored reliable response in AIB to olfactory stimuli. Since AIB is known to promote reversals, the motor-related feedback from RIM to AIB serves as a positive feedback signal, similar to the RIM-dependent input to AIY reported in our study. Since AIB receives input directly from RIM, examining its activity will not directly inform the role of motor representation in AIY. An ideal experiment in this case would be to directly measure voltage dynamics in AIY in WT and RIM-ablated animals. This may be possible when more reliable voltage indicators or electrophysiological techniques become accessible but is outside the scope of this study.

References

1. L. Luo, D. A. Clark, D. Biron, L. Mahadevan, and A. D. T. Samuel. Sensorimotor control during isothermal tracking in *Caenorhabditis elegans*. The Journal of Experimental Biology, 209(3):4652–4662, 2006

2. V. Venkatachalam, N. Ji, X.Wang, C. Clark, J. K. Mitchell, M. Klein, C. J. Tabone, J. Florman, H. Ji, J. Greenwood, A. D. Chisholm, J. Srinivasan, M. Alkema, M. Zhen, and A. D. T. Samuel. Pan-neuronal imaging in roaming *Caenorhabditis elegans*. Proceedings of the National Academy of Sciences of the United States of America, 113(8):E1082-8, 2016

3. M. Chalfie, J. E. Sulston, J. G. White, E. Southgate, J. N. Thomson, and S. Brenner. The neural circuit for touch sensitivity in *Caenorhabditis elegans*. Journal of Neuroscience, 5(4):956–964, 1985. ISSN 02706474. doi: 10.1523/jneurosci.05-04-00956.1985

4. T. Kawano, M. D. Po, S. Gao, G. Leung,W. S. Ryu, and M. Zhen. An imbalancing act: Gap junctions reduce the backward motor circuit activity to bias *C. elegans* for forward locomotion. Neuron, 72(4):572–586, 2011

5. Y. B. Qi, E. J. Garren, X. Shu, R. Y. Tsien, and Y. Jin. Photo-inducible cell ablation in *Caenorhabditis elegans* using the genetically encoded singlet oxygen generating protein miniSOG. Proceedings of the National Academy of Sciences of the United States of America, 109(19):7499–7504, 2012

6. S. Gao, S. A. Guan, A. D. Fouad, J. Meng, T. Kawano, Y. C. Huang, Y. Li, S. Alcaire, W. Hung, Y. Lu, Y. B. Qi, Y. Jin, M. Alkema, C. Fang-Yen, and M. Zhen. Excitatory motor neurons are local oscillators for backward locomotion. eLife, 7:1–32, 2018

7. D. Witvliet, B. Mulcahy, J. K. Mitchell, Y. Meirovitch, D. R. Berger, Y. Wu, Y. Liu, W. X. Koh, R. Parvathala, D. Holmyard, R. L. Schalek, N. Shavit, A. D. Chisholm, J.W. Lichtman, A. D. T. Samuel, and M. Zhen. Connectomes across development reveal principles of brain maturation in *C. elegans*. bioRxiv, 2020

8. S. H. S. H. Strogatz. Nonlinear dynamics and chaos : with applications to physics, biology, chemistry, and engineering. Westview Press, Boulder, CO, 2015. ISBN 0813349109

9. A. Sordillo, C. I. Bargmann, T. Rockefeller, C. Z. Initiative, and R. City. Behavioral control by depolarized and hyperpolarized states of an integrating neuron by. bioRxiv, pages 1–54, 2021. doi: 10.1101/2021.02.19.431690

10. A. L´opez-Cruz, A. Sordillo, N. Pokala, Q. Liu, P. T. McGrath, and C. I. Bargmann. Parallel Multimodal Circuits Control an Innate Foraging Behavior. Neuron, 102(2):407–419.e8, 2019. ISSN 10974199. doi: 10.1016/j.neuron.2019.01.053

11. M. Ikeda, S. Nakano, A. C. Giles, L. Xu, W. S. Costa, A. Gottschalk, and I. Mori. Context-dependent operation of neural circuits underlies a navigation behavior in *Caenorhabditis elegans*. Proceedings of the National Academy of Sciences of the United States of America, 117(11):6178–6188, 2020

12. D. A. Clark, D. Biron, P. Sengupta, A. D. T. Samuel, Clark, D. Biron, P. Sengupta, and A. D. T. Samuel. The AFD sensory neurons encode multiple functions underlying thermotactic behavior in *Caenorhabditis elegans*. Journal of Neuroscience, 26(28):7444–7451, 2006. ISSN 1529-2401. doi: 10.1523/jneurosci.1137-06.2006

13. A. Gordus, N. Pokala, S. Levy, S. W. Flavell, and C. I. Bargmann. Feedback from network states generates variability in a probabilistic olfactory circuit. Cell, 161(2):215–227, 2015